# Community Climate Simulations to assess avoided impacts in 1.5°C and 2°C futures

Benjamin M. Sanderson[1], Yangyang Xu[2], Claudia Tebaldi[1], Michael Wehner[3], Brian O'Neill[1], Alexandra Jahn[4], Angeline G. Pendergrass[1], Flavio Lehner[1], Warren G. Strand[1], Lei Lin[5], Reto Knutti[6,1], and Jean Francois Lamarque[1]

[1]National Center for Atmospheric Research, Boulder, CO, USA
[2]Department of Atmospheric Sciences, Texas A&M University, College Station, TX, USA
[3]Lawrence Berkeley National Lab, CA, USA
[4]Department of Atmospheric and Oceanic Sciences and Institute of Arctic and Alpine Research, University of Colorado, Boulder, CO, USA
[5]School of Atmospheric Sciences, Sun Yat-sen University, Guangzhou, China
[6]Institute for Atmospheric and Climate Science, ETH, Zurich, Switzerland

*Correspondence to:* Benjamin Sanderson (bsander@ucar.edu)

**Abstract.** The Paris Agreement of December 2015 stated a goal to pursue efforts to keep global temperatures below 1.5°C above pre-industrial levels and well below 2°C. The IPCC was charged with assessing climate impacts at these temperature levels, but fully coupled equilibrium climate simulations do not currently exist to inform such assessments. In this study, we produce a set of scenarios using a simple model designed to achieve long term 1.5°C and 2°C temperatures in a stable climate.

These scenarios are then used to produce century scale ensemble simulations using the Community Earth System Model, providing impact-relevant long term climate data for stabilization pathways at 1.5°C and 2°C levels and an overshoot 1.5°C case, which are realized (for the 21st century) in the coupled model and are freely available to the community. Here we describe the design of the simulations and a brief overview of their impact-relevant climate response. Exceedance of historical record temperature occurs with 60 percent greater frequency in the 2°C climate than in a 1.5°C climate aggregated globally, and with twice the frequency in equatorial and arid regions. Extreme precipitation intensity is statistically significantly higher in a 2.0°C climate than a 1.5°C climate in some specific regions (but not all). The model exhibits large differences in the Arctic which is ice-free with a frequency of 1 in 3 years in the 2.0°C scenario, and 1 in 40 years in the 1.5°C scenario. Significance of impact differences with respect to multi-model variability is not assessed.

## 1 Introduction

The Paris Agreement of 2015 changed the landscape of climate negotiations by framing the debate on future policy in terms of temperature targets which would require substantial globally coherent emissions reductions in the near future (Sanderson et al., 2016). Mitigation efforts required to achieve a likely probability of staying below the upper limit (2°C) exceed those combined efforts currently pledged by the countries (Rogelj et al., 2016; Fawcett et al., 2015). The Representative Concentration Pathways (RCPs) for the Coupled Model Intercomparison Project (CMIP) which informed the 5th Assessment Report of the IPCC (AR5,

Pachauri et al. 2014) diverged in 2005. Since then, the world's greenhouse gas emissions have been closer to the highest emissions pathway (RCP8.5) than any other, even accounting for a recent slowdown in emissions growth (Van Vuuren et al., 2011; Quéré et al., 2016) (aerosol evolution differs relatively much less between RCPs, which are so far broadly in line with observations Klimont et al. (2013)). Current policy (if enacted) would result in a middle-of-the-road emissions pathway resulting in a 2.5-4°C warming level (Kitous and Keramidas, 2015). For the lower Paris Agreement temperature goal of 1.5°C, coherent efforts beginning in 2017 would require both emissions rate reductions of at least 5% per year (Sanderson et al., 2016) and likely requiring substantial commitment to negative net carbon emission technologies in the 2nd half of the century (Smith et al., 2016).

Aside from the feasibility or costs of any scenario, it is important to quantify in which ways a climate of 1.5°C above pre-industrial would be different from 2°C, for others to assess whether avoided impacts could justify the costs of the more stringent mitigation. The study of a 1.5°C world, however, is complicated by a number of factors. The lower temperature goal of 1.5°C exhibits less warming than would likely be achieved in the most aggressive mitigation Representative Concentration Pathway (RCP2.6) considered in the $5^{th}$ Assessment Report of the IPCC (Van Vuuren et al., 2011). Although some individual models exhibited less than 1.5°C warming in this scenario (median warming was 1,6C), CESM warming in this scenario is closer to 2 degrees (Meehl et al., 2013). Furthermore, no individual fully-coupled model has performed a large ensemble of RCP2.6 which fully samples internal variability at a low forcing level.

Second, the RCP2.6 scenario diverged from historical emissions pathways in 2005, and the lack of significant global mitigation action since that point means that future mitigation action required to achieve 1.5°C are now radically different from what would have been necessary in 2005 (Sanderson et al., 2016; Stocker, 2013). As such, the transient climate evolution of RCP2.6 could noticeably differ from a 2.0°C scenario where mitigation action begins in 2017.

Numerous strategies have been proposed for addressing this discrepancy of scenarios, comprehensively listed in James et al. (2017). Pattern scaling techniques, which assume that patterns of temperature and precipitation change can be scaled by global mean temperatures can produce quite skillful reproductions of mean climate shifts within 21st century projections (Tebaldi and Arblaster, 2014). However, climate impacts are often functions of the extremes of the distribution, which may not scale in a simple fashion with global mean temperature, especially for precipitation (Pendergrass and Hartmann, 2014). Another approach is to 'time-shift', by taking periods in existing simulations where global mean temperatures equal the warming level of interest. Schleussner et al. (2016) used this approach to compare impacts at 1.5°C and 2°C. However, time-shifting approaches can be complicated when using a period of transient change still exhibiting a strong trend, because the pattern of warming may differ from the equilibrium state (Herger et al., 2015).

An international modelling effort, "Half a degree Additional warming, Prognosis and Projected Impacts" (HAPPI, Mitchell et al. 2017, 2016) has been proposed to fill the gap for simulations to inform the planned IPCC special report on 1.5°C. This effort uses prescribed sea surface temperatures (SSTs) which are consistent with a scaled CMIP5-mean estimate of predicted equilibrium 1.5°C and 2.0°C climates. This approach has advantages; it is computationally cheap so allows for huge ensembles to be run providing samples of extremes and rapidly deployable for a large number of modeling groups and can provide a multi-model assessment of impacts at the two temperature levels referred to in the Paris Agreement. However, there are potential

limitations to the approach. Because simulations in HAPPI will have one of a finite set of predefined SST patterns, the estimate of significance of the difference in climate states will not completely sample ocean-driven variability. In addition, the use of a single 10 year evolution in SSTs will produce a narrow sample of modes of internal climate variability, but a comprehensive assessment would require a thorough sampling of coupled ocean-atmosphere variability.

Here we present an ensemble of transient coupled climate simulations with the Community Earth System Model (CESM,version 1 Hurrell et al. 2013) which achieve the 2°C and 1.5°C goals in line with the Paris targets. These simulations provide a test case for conclusions inferred from other methodologies using the same model (such as pattern scaling, or HAPPI). In this paper, we first document a simple climate model emulator which is able to predict the transient evolution of global mean temperature, and using this emulator we produce concentration scenarios which result in stable 2°C and 1.5°C scenarios in CESM for the

$21^{st}$ century, with a third scenario describing a brief overshoot which returns to 1.5°C by 2100. We then briefly assess the broad climatic features of these two scenarios, including how they differ in both mean state and in the frequency of extremes which might relate most strongly to societal impacts. We aim in this study to provide a short overview of differences in impact-relevant climate variable (a full list of available climate variables is available online (NCAR, 2017)), with the hope that further studies will focus in more detail on specific processes, regions or societal impacts.

## 15  2   Methods

### 2.1   Emulation

The simulations in this study are produced to inform assessment of impacts at 1.5°C or 2°C above pre-industrial levels, and so the end goal is to produce simulations which equilibrate at those temperature targets. Other studies (Sanderson et al., 2016; Meinshausen et al., 2008; Rogelj et al., 2015) have posed the question in terms of devising scenarios which would produce

a given temperature target with a certain likelihood (i.e., a certain frequency among a set of GCMs that respond to forcing differently from one another).

     But the Paris Agreement requested climate information from IPCC on specific temperature levels: 1.5 and 2°C above pre-industrial levels. In a simple model where variability is limited, this can be achieved by interactively adjusting emissions as a function of past warming (Zickfeld et al., 2009). But in the presence of large internal variability, and in order to produce a

GCM simulation which achieves these goals, we need to perform a reverse calibration: to produce an emission scenario which would result in a specific temperature outcome. Because of the computational expense of running a GCM and the natural variability obscuring the forced signal, this inverse problem can alternatively be solved by employing an emulator which can predict the global mean concentration and temperature trajectory for a given emissions scenario, and iterating the emissions scenario parameters such that the desired stable temperature level is achieved.

Some simple climate models already exist in the literature, but we chose to design our emulator as a community project, and as such to have it be open-source and publicly available for further community research in line with the climate model and the climate data that we produce. We have provided a simple multi-gas climate model to perform this emulation, the Minimal Complexity Earth Simulator (MiCES, 2016) written in MATLAB and capable of emulating the forced global mean temperature

and multi-gas concentration evolution of a more complex model with a minimal set of free parameters. The emulator design and calibration process is discussed at length in the supplementary material section A and B, and the code is provided with this manuscript. For the purposes of this study, the model's parameters were calibrated so that MiCES emulates the Community Earth System Model (version 1.1.1, CESM1-CAM5).

CESM1-CAM5 is a single climate model, and like any model is subject to biases in both its present day simulations and in future projections. As such, the ensemble spread in this study does not represent true uncertainty in future projections, rather a single estimate of climate evolution. However, multi-model assessments in the past have indicated that CESM1-CAM5 is one of the better performing models in the CMIP5 archive. Sanderson et al. (2015a) found this model to be the best performing in a selection of mean state metrics, and Flato (2013), Figure 9.37 shows that the model has one of the better simulations of extreme temperature and precipitation metrics in the CMIP5 archive. CESM also has a higher climate sensitivity (4.0K, Gettelman et al. (2012)) than the CMIP5 mean (3.6K, Webb et al. (2013)), and so emissions need to be reduced faster than average for this model in order to meet any given temperature target.

## 2.2 Scenarios

Once the emulator has been calibrated, we design emissions scenarios which are predicted to produce stable 2°C and 1.5°C climates in CESM1-CAM5 (as in Figure 1(b)). We use the methodology established in Sanderson et al. (2016) to define idealized emissions pathways which produce a smooth emissions trajectory from historical trends into a period of emissions reduction, a net-negative emissions phase (where net-negative emissions are constrained to not exceed levels seen in the SSP database(IIASA, 2016)) and a long-term relaxation to levels compatible with a stable global mean temperature. Scenario parameters are adjusted to produce multi-gas emissions scenarios which achieve 3 outcomes relevant to the 1.5 and 2°C goals:

**1.5°C 'never-exceed'** (*1.5degNE*) This scenario is designed such that expected multi-year global mean temperature never exceed 1.5°C above pre-industrial levels in CESM-CAM5 (where pre-industrial is taken as the 1850-1920 mean - averaging over the period before the model initial conditions were branched, and using a 70 year rather than 30 or 50 year mean because there is only one ensemble member for this period). Emissions follow RCP8.5 until 2017, after which carbon emissions rapidly decline reaching 50 percent of 2017 levels by 2027. Combined fossil fuel and land use carbon emissions reach net-zero (carbon neutral) in 2038. CO2 emissions reach a peak net negative level in 2065, with a net flux of -1.8GtC/yr. After this, negative emissions fluxes are reduced, reaching -0.9GtC/yr by 2100 (Figure 1(a)). The magnitude of negative emissions continues to decline throughout the 22nd century, reaching -0.3GtC/yr by 2200 (Figure C1(a)).

**1.5°C 'overshoot'** (*1.5degOS*) This scenario is designed such that expected global mean temperature briefly overshoot before returning to 1.5°C by 2100 in CESM1-CAM5. Emissions follow RCP8.5 until 2017, after which emissions decline slightly less rapidly than in *1.5degNE*, such that emissions are halved from 2017 levels by 2032. In this scenario, combined fossil fuel and land use carbon emissions reach net-zero in 2046. The overshoot requires a larger late century negative emissions commitment, with a peak net negative flux of -4.0GtC/yr in 2080. After this, negative emissions

fluxes are rapidly reduced, reaching -1.0GtC/yr by 2100, but then remain slightly negative throughout the $22^{nd}$ century, reaching -0.5GtC/yr by 2200.

**2.0°C (*2.0degNE*)** This scenario is designed such that expected multi-year global mean temperature never exceed 2°C above pre-industrial levels. Emissions follow RCP8.5 until 2017, after which emissions decline significantly less rapidly than in *1.5degNE*, such that emissions are halved from 2017 levels by 2042. In this scenario, combined fossil fuel and land use carbon emissions reach net-zero in 2078. The scenario still requires a negative emissions phase but much smaller than the other two scenarios, with a peak net negative flux of -0.8GtC/yr in 2120. After this, negative emissions fluxes are slowly reduced, reaching -0.5GtC/yr by 2200.

Using the calibrated MiCES model, the three emissions scenarios are used to produce concentration pathways which can be used in a CESM1-CAM5 simulation. We use CESM1.1.1 with the Community Atmosphere Model (version 5, CAM5) at finite volume 1 degree resolution and the Parallel Ocean Program version 2 at 1 degree resolution, to be consistent with the previous large ensemble studies using RCP8.5 (Kay et al., 2015) and RCP4.5 (Sanderson et al., 2015b). In each of the low emission scenarios in this paper, only well-mixed greenhouse gas concentration are changed between scenarios, all other forcings (land use, aerosol emissions, and ozone) follow RCP8.5 throughout the $21^{st}$ century as in Kay et al. (2015)). A set of 10 simulations are conducted for scenarios *1.5degNE*, and *2.0degNE*, branching from the corresponding historical simulations of Kay et al. (2015) in 2006, running through 2100. A set of 5 simulations are conducted for scenario *1.5degOS*. As such, for each scenario the CESM ensemble samples uncertainty due to internally generated variability conditional on an assumed scenario and model design.

It should be noted that the assumptions of RCP8.5 trajectories for non greenhouse gas forcers is implemented for practical reasons to make the present study tractable. However, a fully self consistent 1.5 degree scenario from an Integrated Assessment Model (IAM) would likely have slight differences. Sulfur emissions are lower in RCP2.6 than in RCP8.5, and maybe lower still in a 1.5 degree scenario (although there large differences in sulfur emissions between individual IAMs with the same policy constraints). However, any given global temperature target could be achieved with different combinations of aerosol forcing and greenhouse gas forcing, but with regional differences in temperature and precipitation (Xu et al., 2015; Pendergrass et al., 2015), and changes in land use necessary for large scale biofuel production would change surface albedo (Caiazzo et al., 2014). An IAM could also have additional degrees of freedom, with the capacity to reduce N2O and CFC emissions below the RCP2.6 minimum levels.

## 3 Temperature, Sea Level, and Sea Ice

### 3.1 Global-scale mean changes

Figures 1(a) and (b) show the CESM ensemble mean global mean temperature trajectory for the three scenarios, which broadly shows that the emulation process was able to correctly predict the expected ensemble mean temperature evolution for CESM (compare with Figure C1 in the Supplemental Material). The *1.5degNE* scenario reaches 1.5°C above pre-industrial levels and

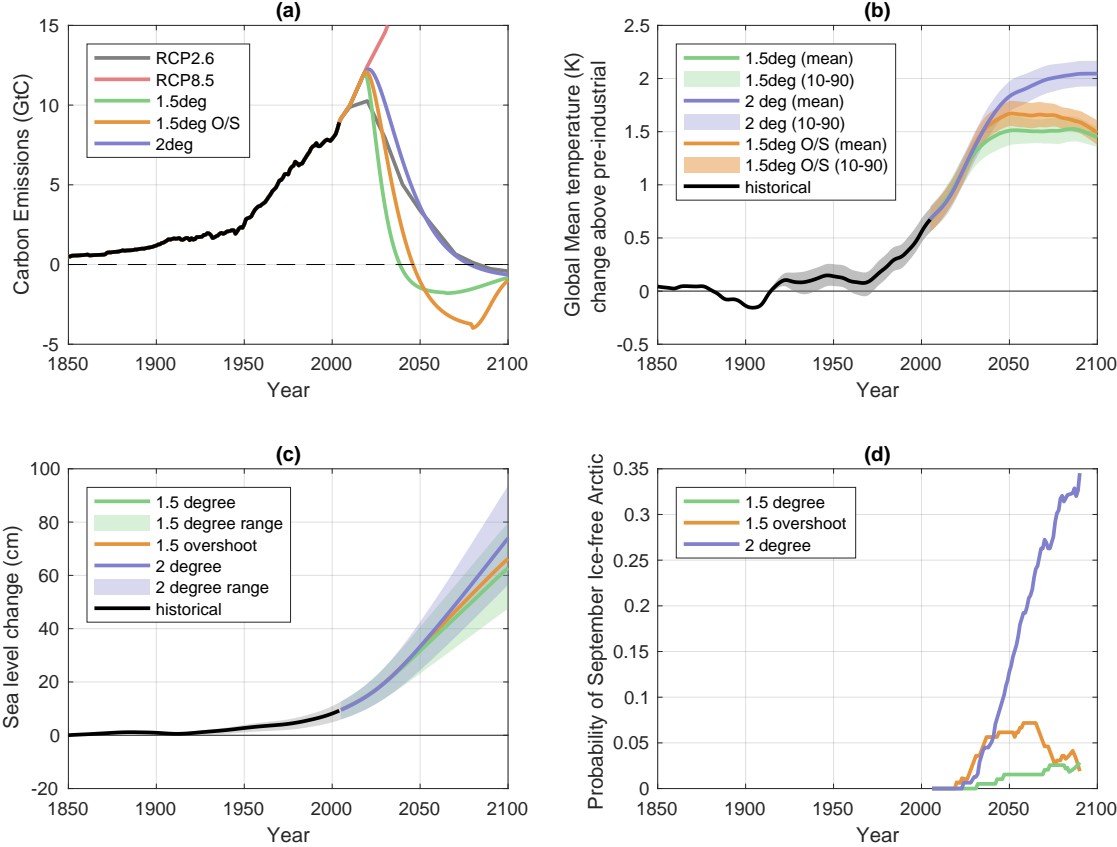

**Figure 1.** Results from the CESM low emissions scenarios. (a) shows the total carbon emissions trajectory (fossil fuel, cement and land use) used to drive the model simulations. (b) shows the range of annual global mean temperatures changes for the three scenarios, 1.5°C never-exceed (green), 1.5°C overshoot (orange) and 2.0°C (purple). Lines are the most likely values, while the shaded areas are the 10-90 percent expected range in the CESM ensemble. (c) shows the corresponding sea level rise using global mean temperature relationships from Kopp et al. (2016) and (d) shows the annual likelihood derived from the CESM ensembles that the Arctic will be ice free in September (20 year running mean).

then maintains this temperature until 2100. The *1.5degOS* scenario reaches a peak temperature in 2050 of 1.7° above pre-industrial before cooling to 1.5°C by 2100, and the *2.0degNE* scenario reaches slightly over 2.1°C by 2100. Note that although these results suggest that the predicted emulation of stable temperatures in the emulator is validated for the evolution of global temperatures in the coupled system in the 21st century, there may be ocean dynamical processes at longer timescales in the coupled model which are not represented in the emulator's thermodynamic ocean. As such, we have not tested the stability of global temperatures at multi-century timescales with these emissions pathways.

We can consider what these global temperature trajectories mean for global scale climate change. Figure 1(c) shows the expected range of possible sea level rise over the $21^{st}$ century for the three scenarios. The CESM1-CAM5 model can only simulate one component of future sea level rise, that of thermal expansion. Notably it does not include ice sheet melt, so in order to show the range of possible sea level trajectories we use formulation of Kopp et al. (2016), using the 10th and 90th percentiles of the posterior parameter distributions for the semi-empirical model fitted to Mann et al. (2009) data to define range. The inherent uncertainty in the sea level response at a given emissions level is greater than the difference between the scenarios considered here. Using these estimates, the *1.5degNE* scenario would result in 50–80cm of sea level rise by the end of the century, while the *2.0degNE* scenario would result in 60–90cm of rise. On a longer timescale, these two scenarios will further diverge. Schaeffer et al. (2012) found that by 2300, the most likely estimate from a 1.5°C stable temperature would be a stable 150cm of sea level rise, but a 2°C stable temperature would result in 270cm and still increasing in 2300.

In our simulations, we find one of the most dramatic divergences between the 1.5 and 2°C simulations comes at high latitudes. This is illustrated in Figure 1(d), which shows the annual likelihood of ice-free Arctic September conditions in the three scenarios. Ice-free is defined as a condition where September average Arctic sea ice area is less than 1 million square kilometers. Our analysis counts the number of ice-free September years in a 20 year moving window in each 10 member ensemble to assess the probability of ice-free conditions as a function of time. We find that in *1.5degNE*, ice-free conditions remain rare, a 1 in 40 year occurrence by the end of the century. In *1.5degOS*, the likelihood of ice-free conditions peaks in 2060, where there is a 1 in 15 chance of ice-free conditions - but this likelihood then declines such that the likelihood is similar to *1.5degNE* by 2100. However, the *2.0degNE* shows significantly greater chances of ice-free conditions; by 2100, 1 in 3 years are simulated as ice-free with likelihoods still increasing at the end of the century. It is notable that the difference between the 2.0°C and 1.5°C scenarios largely arises due to the reduced summer survival of multi-year ice in *2.0degNE*, with only 8 percent of annual mean sea-ice in 2100 in *2.0degNE* that is older than one year, compared with 15–20 percent in the *1.5degNE* and *1.5degOS* (and 35 percent in 2005 - see Figure C2 in the Supplemental Material). Compared to recent statistical analysis of CMIP5 RCP8.5 and RCP4.5 simulations by Screen and Williamson (2017), our results show a higher likelihood of ice-free conditions for *1.5degOS*. The occurrence of ice-free conditions in a 1.5°C world was deemed "exceptionally unlikely" in Screen and Williamson (2017), but occurs in three out of ten *1.5degOS* ensemble members at least once before the end of the 21st century. All of these occur after the first crossing of the 1.5°C threshold defined by Screen and Williamson (2017), illustrating the importance of dedicated stabilization simulations for these targets in order to assess the probability of extreme-value climate events such as an ice-free Arctic in the presence of large internal variability. And while it should be noted that there is large diversity in the rate of loss of Arctic sea ice in CMIP5 models (Mahlstein and Knutti, 2012), the CESM sea-ice loss per degree of global warming to date is less than that which has been observed (Rosenblum and Eisenman, 2017).

## 3.2 Regional Differences

The regional differences in mean climate between the two scenarios is illustrated in Figure 2. Spatial patterns of warming remain similar in all simulations, with land regions, especially in high latitudes and desert regions, warming more than average. It is notable that for a given global mean temperature target, land temperature warming will be greater: the ensemble mean

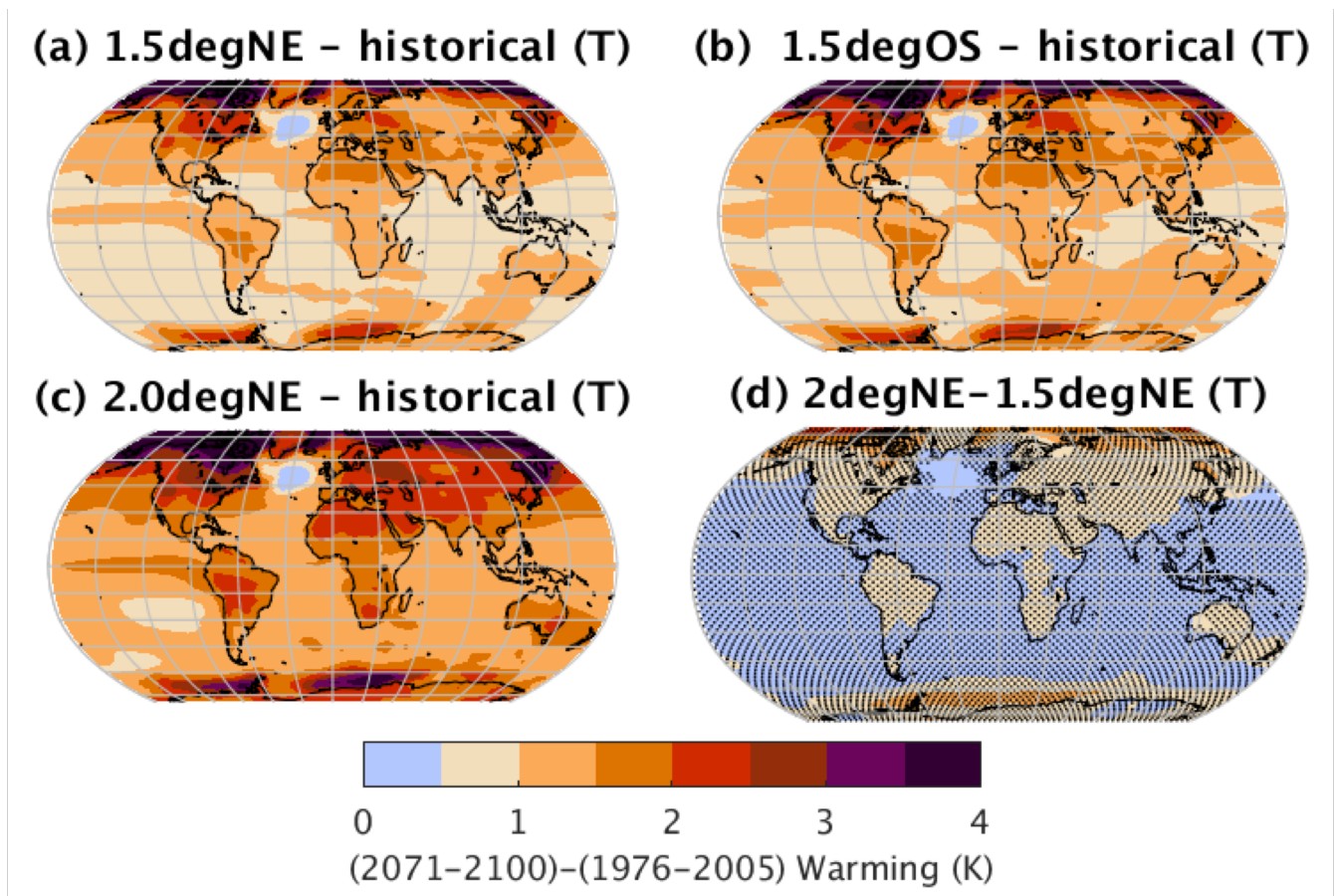

**Figure 2.** Maps showing ensemble mean 2071-2100 temperature changes relative to 1976-2005 historical conditions in *1.5degNE* (a), *1.5degOS* (b), *2.0degNE* (c) scenarios. Subplot (d) shows the difference between mean 2080-2100 conditions in *1.5degNE* and *2.0degNE*, where significant regions are stippled. Significance is defined as a pixel in which the difference between the mean of the *2.0degNE* and *1.5degNE* ensembles exceeds the standard deviation of 2080-2100 values in the *2.0degNE* ensemble.

2071-2100 land surface temperature is 1.8°C warmer than 1851-1880 in *1.5degNE*, and 2.4°C warmer in *2.0degNE*. At high latitudes, Greenland, for example has an ensemble mean warming of 2.3°C in *1.5degNE* and 3.1°C in *2.0degNE*.

In almost all regions of the globe, the difference in 30 year mean warming for 2071-2100 between *2.0degNE* and *1.5degNE* is statistically significant compared with intra-ensemble random variability (shown in Figure 2(d)). The differences in mean temperature change between *1.5degNE* and *1.5degOS* are subtle, the latter showing more extensive ocean warming and slightly more warming over North America and Central Asia - but most regions do not show detectable differences between the two scenarios at the end of the 21$^{st}$ century (not shown).

### 3.3 Temperature Extremes

Human impacts such as mortality vary disproportionately with the upper tail of the distribution of temperature (e.g. Guo et al. (2014); Gasparrini et al. (2016)), hence it is of relevance to consider the frequency with which historical temperature extremes will be exceeded in the future. We consider this in a couple of ways, firstly by adapting the methodology of Lehner et al. (2016), which assesses the frequency with which records are exceeded in the future. Figure 3(a) shows the fraction of global

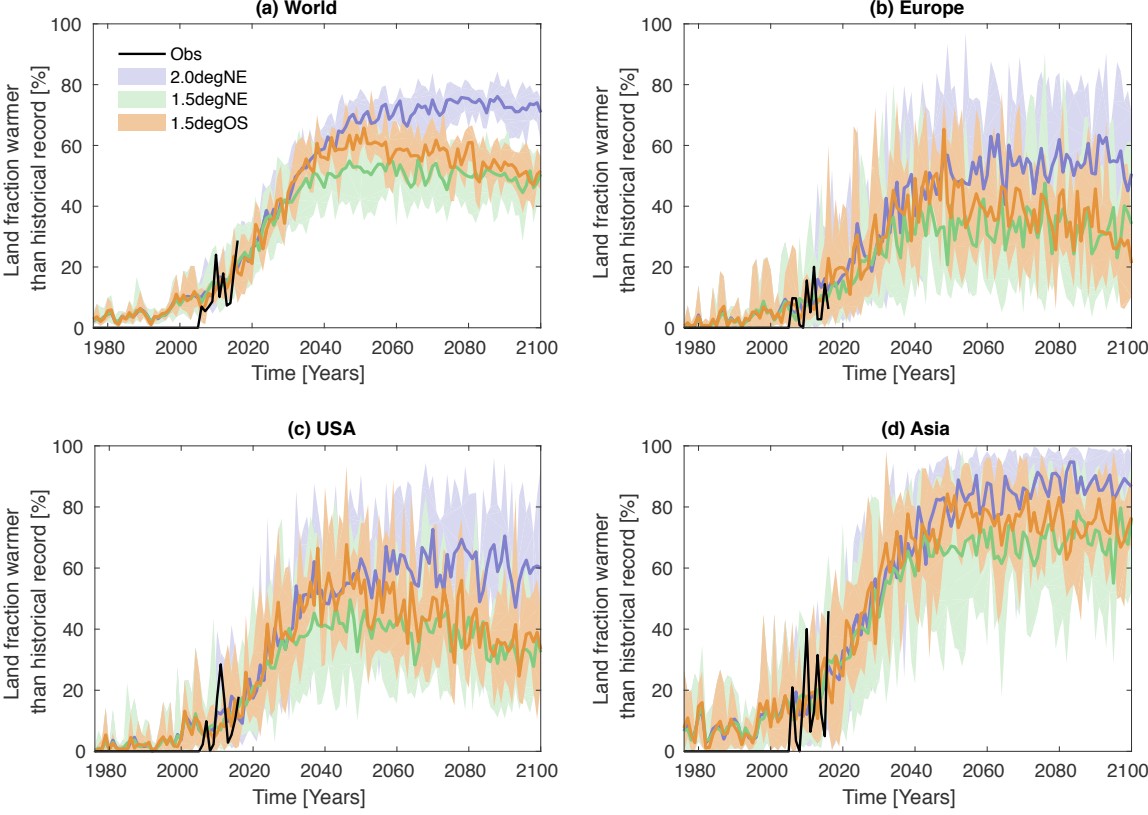

**Figure 3.** Fraction of land in which the observed historical summer temperature record defined during the period 1976-2005 is broken in any given year. Summer is defined for each grid cell as the climatologically warmest 3-month period. (a) is the fraction of global land area, while (b-d) refer to the fraction of a specific region. The solid central line is the ensemble mean for each scenario, and the shaded areas represent the full ensemble range.

land area which is simulated to exceed the observed 1976-2005 record summer temperature in any given year, for each of the three scenarios. By 2040, there is little separation between the scenarios - globally, about 50-60 percent of the global land area exceeds historical temperatures in any given year. However, towards the end of the century, there is a significant difference.

45-55 percent of land area is simulated to exceed the historical record in *1.5degNE*, compared with 70-80 percent in *2.0degNE*. Observations are from Rohde et al. 2013, as in Lehner et al. (2016).

It is notable that some records in Figure 3 are exceeded before 2005 because the historical evolution of the CESM ensemble differs from the real-world historical evolution and there could potentially be some model regional model biases. However, the behaviour of CESM in the period 2006-2016 is in within the range of model record exceedance (both globally and in each of the regions considered), giving confidence that regional biases are not strongly influencing this metric. Note that if the records were taken from the historical simulations of CESM itself for consistency, almost all historical records would be a higher temperature because the effective sample period in a 10 member ensemble is 300 years for the period 1976-2005, which causes a 30 percent reduction in end of 21st century record exceedance (see discussion with reviewer M.Sarofim for further details).

Assessed at a regional level, there is greater ensemble spread in the fraction of land area which exceeds the historical record summer temperature (as would be expected from the scale/variability relationships discussed in Deser et al. 2012). However, in the US and in Europe, roughly double the fraction of land experiences historical record breaking summers in *2.0degNE* relative to *1.5degNE* by 2071-2100. Over Asia, the ratio of warming to natural variability is greater for all scenarios, so even the *1.5degNE* scenario is simulated to exceed historical records in 50-90 percent of the region each year (compared with 80-100 percent in *2.0degNE*).

We can consider the spatial distribution of extreme temperature events by following the methodology of Tebaldi and Wehner (2016), which fits a Generalized Extreme Value (GEV) distribution to the upper tail of 3-day temperature averages for both the historical period (1976-2005 in this case) and the future (2071-2100) under different scenarios. The GEV is used to assess, historically, the 3 day average temperature which would be considered a 1 in 20 year event. Figure 5 shows the increase in the expected temperature for a 1 in 20 year 3-day warm event in each of the scenarios. When compared with Figure 2, it becomes clear that some regions experience a greater increase in extreme temperatures than the mean temperature increase (in *2.0degNE*, 9% of the land area experiences a warming of extremes more than 50% faster than the mean, while less than 1% experiences a warming 50% slower than the mean).

For example, comparing Figure 2(c) and 5(a) - certain regions such as the northern USA/Southern Canada, northern Europe, Eastern China and western Brazil/Bolivia which experience increases in extreme 3-day temperatures of about 4K - about twice the mean temperature increase. Comparing *1.5degNE* and *2.0degNE*, most regions globally show a difference of less than 1°C between the scenarios - with the exception of Canada and Northern Europe which show differences of 1-2°C.

We can also consider the expected frequency of exceedence of historical 1 in 20 year 3-day warm event levels. Figure 4 shows the frequency with which these current temperature extremes would be exceeded in *2.0degNE* and *1.5degNE*. This metric shows the greatest increase in regions with a large ratio of mean change to ensemble variability. Because internal variability is greatest at high latitudes in this model (Kay et al., 2015) and indeed in most CMIP5 models (Mahlstein et al., 2011), the greatest signal to noise is seen at lower latitudes (although the absolute magnitudec of warming is smaller). Central Africa, the Saudi Arabian peninsula, Brazil, Peru and to a lesser extent Western India and the Western USA all show significant increases in the frequency of expected exceedance of the current 1 in 20 year maximum 3-day temperature event. In *1.5degNE*,

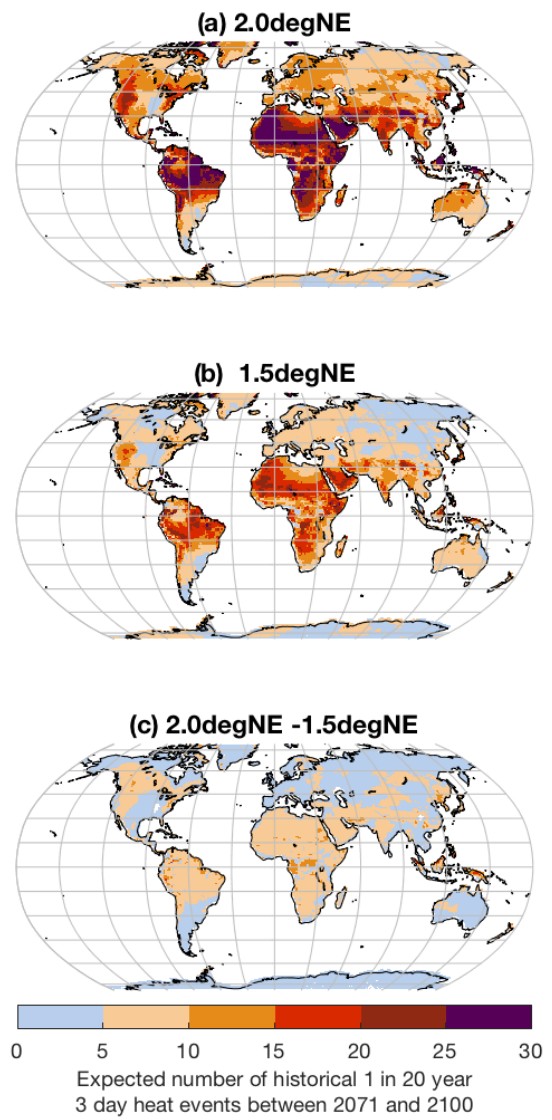

**Figure 4.** Maps showing the expected number of times that the modeled historical 1 in 20 year 3-day warm event in the period 1976-2005 would be exceeded during the period 2071-2100 for (a) *1.5degNE*, (b) *2.0degNE* and (c) the difference between the number of events in each scenario. By definition, no change from historical climate would correspond to a value of 1.5 events per 30 years.

these regions are simulated to experience 10-20 such events between 2071 and 2100, while in *2.0degNE* exceedance would be an annual event for most of the regions in question.

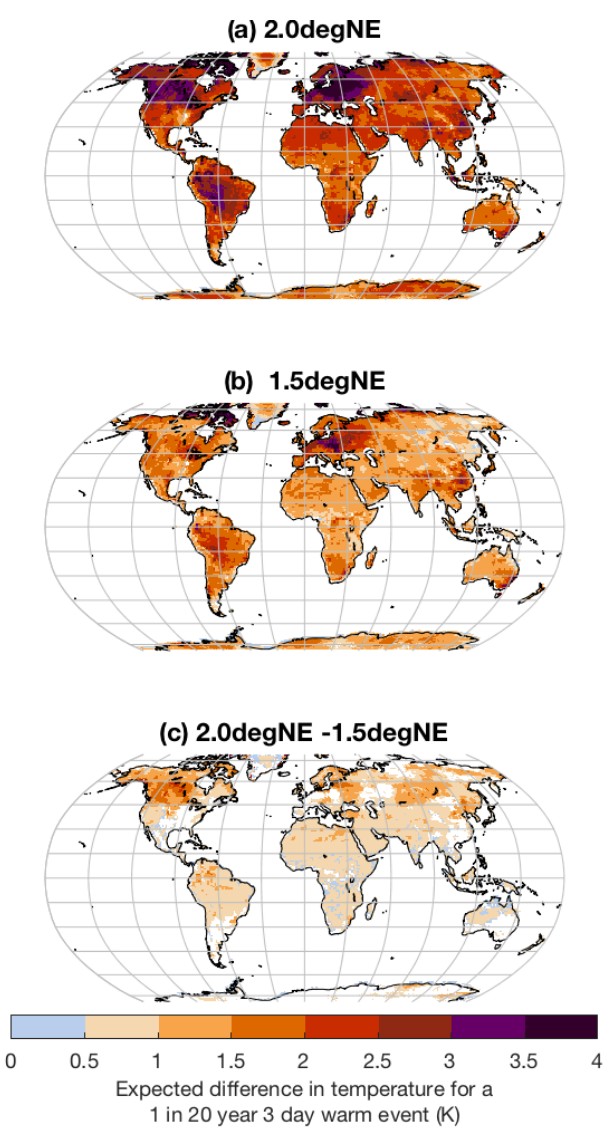

**Figure 5.** Maps showing the expected difference from 1976-2005 in temperature for a 1 in 20 year 3 day warm event for (a) *1.5degNE*, (b) *2.0degNE* and (c) the difference in temperature between each scenario.

## 4 Precipitation changes

### 4.1 Mean

At the global level, there are statistically significant differences in precipitation between 1.5°C and 2°C climates. Global-mean precipitation increases by 3.3 % in *1.5degNE* but 4.5 % in *2.0degNE* from 1976-2005 to 2070-2100 (Figure 6(a); the difference

between the two scenarios exceeds twice the standard deviation of the *2.0degNE*). Over land (Figure 6(b)), mean precipitation increases are slightly less than global mean increases in both cases: 2.8% for *1.5degNE* and 4.3% *2.0degNE* over 1976-2005 levels. At higher latitudes, mean increases are greater at 5.4 and 6.8%. The difference in the mean aggregated land-based precipitation in *1.5degNE* and *2.0degNE* is not significant by the end of the century (neither over all land, nor land north of 20°N).

Precipitation differences between the climate states are less significant at the grid point level (Figure 7). All simulations show increased precipitation in the tropics, decreased precipitation in the subtropics and increased precipitation at high latitudes. Very few land regions show a net decrease in precipitation, but there are some exceptions: the southwest US, Amazonia and Indonesia all suggest reductions in precipitation in *2.0degNE*. In most cases, local differences in precipitation between *1.5degNE* and *2.0degNE* are not statistically significant over land; only over the Arctic and Southern oceans are there significant differences (Figure 7(g)).

## 4.2  Water availability and aridity

Water availability is controlled by joint changes in precipitation minus combined evaporation and transpiration from the surface (hereafter P-E). Figure 7 also shows how P-E changes in the different scenarios. Notably, although very few land regions show a decrease in precipitation, most land regions in all scenarios show a net decrease in P-E by 2071-2100. However, very few regions show a significant difference between changes in *1.5degNE* and *2.0degNE* (exceptions being central Amazonia and the Mediterranean).

We can also consider *aridity*, the ratio of precipitation to potential evapotranspiration (PET, the evaporative demand of the atmosphere; Fu and Feng 2014; Lin et al. 2015). Regions where the ratio of P/PET is less than 0.65 are classified as 'dry land' as in Lin et al. (2015). In these simulations, the increase in dry land in a 2°C world is double that of a 1.5°C world: figure 8(a) shows that the global area of dry land increases by 1.25 million $km^2$ between 1976-2005 and 2071-2100 under *2.0degNE* while under *1.5degNE*, global aridity peaks in 2040, such that there is an increase of only 0.66 million $km^2$ by 2071-2100.

Although the difference in change in total arid area is significant, aggregating changes in aridity by region (Figure 8(b)) suggests that only some regions exhibit statistically significant differences in changes between 2 and 1.5°C climates. Notably both Europe and Southern Africa show large differences in the expected change in aridity by 2071-2100 in *2.0degNE* and *1.5degNE*. Comparing with Figure 7(h), in this is driven largely by Mediterranean regions in Europe and by eastern coast of South Africa and Namibia in Africa.

## 4.3  Extreme Precipitation

The changing intensity of extreme precipitation is an impact-relevant quantity for ecosystems (Knapp et al., 2008), disease transmission (Curriero et al., 2001) and infrastructure planning (Hossain et al., 2010). We consider the changing patterns of extreme precipitation both at the point level and regionally. Figure 9 shows that only some specific regions would expect large increases in extreme precipitation (quantified here as the annual 1 day maximum rainfall) - eastern Amazonia, Congo, Peru,

central India and northern Canada are simulated to experience a 25-50 percent increase. Differences between the *2.0degNE* and *1.5degNE* are subtle by this metric, with very few regions showing significant differences at the grid point level.

However, when precipitation is aggregated over larger regions, significant differences in extreme behavior between scenarios can become apparent (Fischer and Knutti, 2015). This is illustrated in Figure 10, showing the aggregated relative frequency of events which exceed the historical 99th percentile of daily precipitation. Globally aggregated, there is a large and significant difference between the scenarios: a 7-8 percent increase in events above the historical 99th percentile of daily precipitation in *1.5degNE*, and a 13-15 percent increase in *2.0degNE*. When events are aggregated to the regional level, there are still significant differences in some regions between the frequency of events in *2.0degNE* and *1.5degNE*. High northern latitude regions show consistent separation (Alaska, N.Canada, Greenland, N.Asia), as does the Eastern USA, Eastern Africa, Eastern Asia.

In short, at the grid cell level, there is no significant difference in the likelihood of extreme precipitation events in a 2°C and 1.5°C climate. However, there are significant differences at the regionally and globally aggregated scales.

## 5 Conclusions and Discussion

In this study, we investigate whether there are significant differences between 1.5°C and 2.0°C of warming in the physical climate system that might have societal impacts. We present a climate model emulator that solves for a concentration pathway leading to a particular temperature target for a particular climate model. Using this emulator, we design three ensembles of coupled climate simulations with a single climate model, CESM, which produce stable equilibrium global mean temperature at 1.5°C and 2°C above pre-industrial levels, the first set of coupled simulations to our knowledge.

We use these simulations to compare impacts at the two stabilization targets. The question of how substantial this difference in mean climate might be depends on both scale and variable. Mean temperatures are significantly different throughout the globe in a 1.5 and 2°C warmer climate, and various aspects of extreme temperature exposure are distinguishable, especially when aggregated over large areas. There are significant differences in exposure to events exceeding historical heatwaves in a 1.5 and 2°C scenario.

We find also that the difference between 1.5°C and 2°C spans a threshold for the persistence of summer sea-ice in the Arctic, such that ice-free Septembers in the Arctic remain rare in a 1.5°C world, but occur regularly in a 2°C world.

For both mean and extreme precipitation changes, gridpoint-level differences between a 1.5°C and 2°C climate are not significant over most land areas. But when aggregated regionally, some impact-relevant differences become apparent. Most regions experience an increase in the frequency of events above the historical $99^{th}$ percentile of precipitation, and this increase is significantly greater in the 2°C simulations than in the 1.5°C simulations - an effect particularly pronounced in high latitude regions.

Taken together, it can be argued that if these simulations are representative of the true climate system, there could be substantial differences in impact-relevant aspects of climate, at least regionally, between a climate system stabilized at the 1.5°C and 2°C levels. Further targeted study would be necessary to demonstrate how these differences in climate translate to human and ecosystem impacts. Furthermore, the statements of statistical significance in this study account only for the model's

internally variability, and does not account for structural uncertainty. Repeating these experiments with a range of coupled climate models would therefore be of value. The climate model emulator presented here can in principle be applied to other climate models to enable these experiments.

A key factor determining whether or not there are significant differences between 1.5°C and 2°C of warming is the magnitude of the climate system response to a difference of half a degree of warming relative to internal climate variability. This question can be informed by ensembles of coupled climate model simulations including estimates of internal variability arising from the ocean and atmosphere, and yet. These experiments should hopefully provide context (for a single model) for the primary multi-model effort being pursued by the community to address the difference between 1.5°C and 2°C of warming (HAPPI, Mitchell et al. (2017)) which relies on AMIP simulations with prescribed SSTs.

Our study considered two mechanisms to achieve 1.5°C, one which stabilizes by the mid 21st century, while the other overshoots reaching 1.7°C in 2050 and stabilizes at 1.5°C by 2100. Although the focus of the impact studies considered here have compared the equilibrium states at 1.5 and 2°C, our scenarios allow a consideration of additional impacts which the overshoot would imply. In 2050, an additional 10 percent of global land area would be expected to exceed historical summer temperature records in the 1.5°C overshoot, compared with the 1.5°C stabilization case - although differences are not significant at the gridcell level. Our results do not suggest significant differences in sea level rise between the 1.5°C overshoot case and the stabilization case and ice-free Arctic summers are simulated to be rare in both scenarios. Our analysis not suggest any evidence of long term climatic difference post-2100 of the overshoot relative to the stabilization case.

The question of whether a 1.5°C world is 'worth' the mitigation costs beyond the already aggressive goal of 2°C is a complex one, ideally requiring a comprehensive assessment of both the costs and the impacts associated with the two temperature targets. This study focuses on the differences of the physical climate system in order to inform one aspect of this question: are the differences between these temperature targets significant in terms of aspects of climatology which might have societal impacts? To assess the impacts themselves is left for future study, for which we hope these experiments will prove useful.

Our 1.5°C scenarios require net zero emissions by 2040 if the expectation value for global mean temperatures is never allowed to exceed 1.5°C, and by 2045 if a brief overshoot is allowed. These emissions scenarios are conditioned on CESM representing the true climate system, but these dates are broadly consistent with a more general treatment is considered in Sanderson et al. (2016). However, our model contains no representation of society or economy, and cannot be used to assess the more general question of whether this rate of decarbonization is an achievable goal in reality. Rogelj et al. (2013) considered the probabilistic cost estimates of mitigating to different equilibrium targets, finding that a 1.5°C solution was technically possible in some integrated assessment models with assumptions of low future energy demand. However, a number of studies suggest that the window for a plausible 1.5°C future is rapidly closing, if it has not closed already. Huntingford and Mercado (2016) points out that with many current GCMs, present day concentrations would already result in equilibrium global mean temperatures which would be warmer than 1.5°C. As such, mitigation costs are highly sensitive to political inaction, such that by 2020 in Rogelj et al. (2013), there is no level of carbon taxation which could achieve a 50 percent chance 1.5°C climate.

Mitigation to 1.5°C may or may not be politically feasible. However, it is perhaps more useful to think of a 1.5°C climate as the lowest feasible level of warming which could be achieved this century in ideal conditions: assuming low energy demand, a

cooperative global political commitment to decarbonization, fast growth in low carbon technology and development of wide-scale negative emissions infrastructure by mid-century. Simulations such as those presented in this study and those to be conducted in the HAPPI framework provide a quantitative context for the consequences of these idealized conditions failing to be realized. Irrespective of feasibility, these simulations indicate that a relaxation of ambition from the 1.5°C to the 2°C level would result in significantly greater impacts in some regions, at least compared with internal variability in CESM. Further study should consider these results in a multi-model context, using HAPPI and pattern scaling work together with these coupled single model experiments to produce a comprehensive assessment of avoided impacts in high mitigation scenarios.

## 6  Code Availability

The simple model used to create and optimize the scenarios used in this study is open-source and freely available at the MiCES github repository (`https://github.com/benmsanderson/mices`).

## 7  Data Availability

The output data from the simulations produced for this study are freely available at (`http://www.cesm.ucar.edu/experiments/1.5-2.0-targets.html`.

## Appendix A:  Minimal Complexity Earth Simulator (MiCES)

This study uses a simple model emulator developed by the authors in MATLAB to predict the global mean response of CESM to an emissions scenario, a simple energy balance representation of the Earth's temperature and carbon cycle response. The core of the model is a set of five differential equations. The first describes the evolution of the atmospheric temperature:

$$\frac{\partial T_a}{\partial t}(t) = \frac{1}{\kappa_l}(6.3 \cdot log(C_a(t)/C_a(0)) + F(t) - \lambda \cdot T_a(t)) - D_o \cdot (T_a - T_o) \tag{A1}$$

where $t$ is time, $T_a$ and $T_o$ are the atmospheric and ocean temperatures, $F(t)$ is the sum of all non-$CO_2$ forcing, $\kappa_l$ is the thermal heat capacity of the land surface, $C_a$ is the atmospheric carbon content in Pg, $\lambda$ the climate sensitivity parameter in $(K^{-1})$ and $D_o$ is the thermal diffusion-like coupling parameter between the atmosphere and the shallow ocean.

The second equation describes the evolution of the atmospheric carbon content:

$$\frac{\partial C_a}{\partial t}(t) = \frac{E(t) - (\gamma_l + \gamma_o) \cdot \frac{\partial T_a}{\partial t}(t) \cdot (1 + \delta T_a)}{1 + \alpha(\beta_l)} - \beta_o(\alpha C_a - \rho_o C_o), \tag{A2}$$

where $E(t)$ is the carbon emissions at time $t$ in Pg, $\gamma_l$ and $\gamma_o$ are the land and ocean temperature driven carbon feedbacks (in Pg/K), $\delta$ is a carbon feedback amplification factor (allowing the carbon release from the land or ocean to respond non-linearly to temperature, in $K^{-1}$), $\alpha$ is a conversion factor from Pg to atmospheric carbon concentration, $\beta_l$ is the $CO_2$ fertilization

parameter and $\beta_o$ is the carbon diffusion coefficient between the atmosphere and ocean, $(\alpha C_a - \rho_o C_o)$ is the difference in carbon concentration between the atmosphere and ocean, where $\rho_o$ is the conversion factor between ocean carbon content in Pg and ocean carbon concentration. Note that the land carbon cycle feedback is not a function of the land carbon pool in this version, but is modeled using the parameters of Friedlingstein et al. (2003), where land carbon uptake is governed linearly by

atmospheric CO2 content and temperature.

The third equation describes the evolution of the shallow ocean carbon content:

$$\frac{\partial C_o}{\partial t}(t) = \beta_o(\alpha C_a - \rho_o C_o) + \gamma_o(1 + \delta T_a)\frac{\partial T_a}{\partial t}(t) - \beta_{od} \cdot (\rho_o \cdot C_o - \rho_{od} \cdot C_{od}), \tag{A3}$$

where $\beta_o(\alpha C_a - \rho_o C_o)$ is the diffusion-based carbon flux from the atmosphere, $\gamma_o(1 + \delta T_a)\frac{\partial T_a}{\partial t}(t)$ is the loss of carbon from the ocean to the atmosphere due to temperature-driven feedbacks, $\beta_{od}$ is the carbon diffusion coefficient between ocean

and deep ocean and $(\rho_o \cdot C_o - \rho_{od} \cdot C_{od})$ is the carbon concentration difference between the shallow and deep oceans ($C_{od}$ is the conversion factor between deep ocean carbon content in Pg and deep ocean carbon concentration).

The fourth equation describes the evolution of carbon in the deep ocean, $C_{od}$, which changes due to a diffusive flux of carbon from the shallow ocean:

$$\frac{\partial C_{od}}{\partial t}(t) = \beta_{od}(rho \cdot C_o - rho2 \cdot C_{od}), \tag{A4}$$

Finally, the fifth equation describes the temperature evolution of the shallow ocean, where $\kappa_o$ is the thermal heat capacity of the shallow ocean and $D_o$ is the diffusion coefficient. Heat diffusion across the shallow ocean bottom boundary was not found to be a necessary complexity for emulation.

$$\frac{\partial T_o}{\partial t}(t) = D_o \cdot \kappa_o \cdot (T_a - T_o), \tag{A5}$$

The source code for MiCES is included in the supplementary material of this paper. Non-$CO_2$ forcings $F(t)$ are calculated

using the atmospheric chemistry model defined and published in Prather et al. (2012), which calculates the lifetimes and radiative forcings of non-$CO_2$ atmospheric components ($CH_4$, $N_2O$, HCFCs, CFCs), the model includes some bias correction for present day concentrations and growth rates. Aerosols are calculated by assuming that net aerosol radiative forcing is proportional to global, annual $SO_x$ emissions. The coupled equations are solved using the ODE solver (ode45) in MATLAB.

## Appendix B:  Tuning for model emulation

The model defined in Section A has a number of free parameters, both in the climate and carbon cycle equations defined in the text and in the chemistry treatment of Prather et al. (2012). Our aim in this study is to reproduce the behavior of the CMIP5 configuration of CESM1-CAM5. (Hurrell et al., 2013). This requires reproducing the carbon-cycle and atmospheric chemistry behavior of the closed-source MAGICC model used to produce the concentration pathways for the RCPs (Meinshausen et al., 2008), as well as emulating the global mean temperature response of CESM1-CAM5.

| Submodel | Description | Name | Lower | Upper | Calibrated |
|---|---|---|---|---|---|
| climate | Climate Sensitivity ($Wm^{-2}K^{-1}$) | $\lambda$ | 0.6 | 3.8 | 0.9 |
| climate | Land Surface Heat Capacity $Ka^{-1}(Wm^{-2})^{-1}$ | $\kappa_l$ | 1 | 30 | 9.3 |
| climate | Ocean Heat capacity $a^{-1}$ | $D_o$ | 1 | 300 | 21 |
| climate | Atmosphere-ocean diffusion coefficient $Ka^{-1}(Wm^{-2})^{-1}$ | $\kappa_o$ | 0.001 | 1 | 0.21 |
| climate | Pre-industrial $CO_2$ concentration (ppm) | $ppm\_1850$ | 270 | 290 | 287 |
| climate | biosphere $CO_2$ fertilization parameter (Pg/ppm) | $\beta_l$ | 0 | 3 | 1.9 |
| climate | biosphere temperature response (Pg/K) | $\gamma_l$ | -.5 | 0.5 | -.13 |
| climate | Shallow ocean initial carbon stock (Pg) | $C_o(0)$ | 100 | 600 | 600 |
| climate | Ocean carbon diffusion parameter (Pg/ppm) | $\beta_o$ | 0 | 5 | 1.6 |
| climate | Ocean carbon solubility response (Pg/K) | $\gamma_o$ | 0 | 0.5 | 0.2 |
| climate | Deep-shallow ocean carbon diffusion coefficient (Pg/ppm) | $\beta_{od}$ | 0. | 4 | 0.5 |
| climate | Deep ocean initial carbon stock (Pg) | $C_{od}(0)$ | 20000 | 100000 | 100000 |
| climate | Aerosol 1990 forcing ($Wm^{-2}$) | $f_{1990}$ | -3 | 0 | -1.2 |
| CH4 | Present-day OH feedback, unitless | sOH | -0.37 | -0.27 | -0.28 |
| CH4 | Present-day loss frequency due to tropospheric Cl, 1/a | kCl | 0.0025 | 0.0075 | 0.005 |
| CH4 | Present-day loss frequency due to all stratospheric processes, 1/a | kStrat | 0.0073 | 0.02 | 0.01 |
| CH4 | Present-day loss frequency due to surface deposition, 1/a | kSurf | 4e-3 | 8e-2 | 3e-2 |
| CH4 | Present-day loss frequency due to surface deposition, 1/a | cPI | 650 | 750 | 667 |
| CH4 | Pre-industrial concentration, ppb | cPD | 1600 | 1815 | 1670 |
| CH4 | Present-day growth rate, ppb/a | dcdt | 4 | 6 | 4.5 |
| CH4 | Ratio of PI/PD natural emissions, unitless | anPI | 0.8 | 1.2 | 0.98 |
| CH4 | Ratio of PI/PD loss frequency due to tropospheric OH, unitless | aPI | 0.75 | 3 | 1.38 |
| CH4 | Ratio of y2100/PD loss frequency due to tropospheric OH, unitless | a2100 | 0.5 | 2 | 1.09 |
| CH4 | Present-day methyl-chloroform decay frequency, 1/a | kMCF | 0.176 | 0.3 | 0.23 |
| CH4 | Present-day MCF loss frequency due to ocean uptake, 1/a | kMCFocean | -7.1e-3 | 7.1e-3 | 1e-3 |
| CH4 | k(Species+OH)/k(MCF+OH) at 272 K | r272 | 0.1 | 0.66 | 0.12 |
| N2O | Present-day concentration | cPD | 300 | 326 | 310 |
| N2O | Present-day growth rate, ppb/a | dcdt | 0.6 | 0.9 | 0.70 |
| N2O | Pre-industrial concentration, ppb | cPI | 100 | 300 | 294 |
| N2O | Present-day loss frequency due to all stratospheric processes, 1/a | kStrat | 7.e-3 | 9e-3 | 8e-3 |
| N2O | Ratio of PI/PD loss frequency due to stratosphere, unitless | aPIstrat | 0.75 | 1 | 0.80 |
| N2O | Present-day stratosphere feedback, unitless (dln(kStrat)/dln(C)) | sStrat | 0.06 | 0.2 | 0.16 |
| N2O | Conversion between tropospheric abundance and total burden, Tg/ppb | b | 4.78 | 4.9 | 4.83 |
| N2O | MCF fill factor, global/troposphere mean mixing ratios, unitless | fill | 0.96 | 1.0 | 0.98 |
| N2O | Ratio of y2100/PD loss frequency due to stratosphere, unitless | a2100strat | 0.9 | 1.1 | 1.07 |
| N2O | Ratio of y2100/PD loss frequency due to tropospheric OH, unitless | a2100 | 0.9 | 1.1 | 0.99 |
| N2O | Ratio of PI/PD loss frequency due to tropospheric OH, unitless | aPI | 0.5 | 3.0 | 2.23 |

**Table B1.** Parameter Ranges for the MiCES model used in this study, and calibrated parameters for CESM emulation.

The free parameters in the model, as well as their prior boundary values, are listed in Table B1.

In order to tune the model to CESM1-CAM5, we perform historical and future climate simulations which can be compared to CESM1-CAM5. Climate simulations are conducted from 1850 to 2100 for the 3 scenarios. The models are validated against CMIP5 simulations from CESM-CAM5. The inputs to MiCES are global total emissions of greenhouse gas emissions ($CO_2$, $CH_4$, $N_2O$, CFCs, HCFCs, CO). There is no explicit aerosol scheme in MiCES, instead global mean $SO_2$ emissions are used as a scaling factor for net anthropogenic aerosol forcing.

The CMIP5 greenhouse gas pathways are derived from MAGICC (Meinshausen et al., 2008), hence our calibration target is actually a hybrid of CESM and MAGICC, where temperatures are matched to CESM and greenhouse gas concentrations to MAGICC's CMIP5 default configuration.

The methane and nitrous oxide components are each calibrated in isolation because their evolution is not sensitive to the temperature or carbon state of the model. In each case, an error function is computed as the sum-squared difference in concentration over the integration period for the 3 scenarios:

$$E^{N_2O}(p) = \sum_{s=1}^{3} \sum_{t=1850}^{2100} (c_{mices}^{N_2O}(t,s,p) - c_{magicc}^{N_2O}(t,s))^2 \tag{B1}$$

$$E^{CH_4}(p) = \sum_{s=1}^{3} \sum_{t=1850}^{2100} (c_{mices}^{CH_4}(t,s,p) - c_{magicc}^{CH_4}(t,s))^2 \tag{B2}$$

where $E^{N_2O}(p)$ and $E^{CH_4}(p)$ are the error terms for parameter state $p$, $c$ is the annual mean concentration of methane or nitrous oxide at time $t$ in MiCES and MAGICC, $s$ represents the 3 scenarios (RCP2.6, RCP4.5 and RCP8.5). The parameter state $p$ is initialized from the default values in Prather et al. (2012) and optimized to minimize the value of $E^{CH_4}(p)$ using MATLAB's 'fmincon' function. Other non-$CO_2$ components (CFCs, HCFCs, CO) are simple decay functions kept at the default values from Prather et al. (2012).

Once the non-$CO_2$ components are calibrated, the climate component is tuned by considering the temperature from CESM1-CAM5 and the $CO_2$ trajectories from MAGICC from the CMIP5 RCP2.6, RCP4.5 and RCP8.5 simulations. We start by performing a $10^6$ member perturbed ensemble of the model for 3 different scenarios - RCP2.6, RCP4.5 and RCP8.5. Each parameter in the 'climate' sub-component in Table B1 is perturbed using a flat prior sampling with lower and upper bounds as listed. A cost function $E^{climate}(p)$ for the climate components is calculated as a product of combination of the $CO_2$ and temperature cost functions:

$$E^{CO_2}(p) = \sum_{s=1}^{3} \sum_{t=1850}^{2100} (c_{mices}^{CO_2}(t,s,p) - c_{magicc}^{CO_2}(t,s))^2 \tag{B3}$$

$$E^{T^g}(p) = \sum_{s=1}^{3} \sum_{t=1850}^{2100} (T_{mices}^{T^g}(t,s,p) - c_{CESM-CAM5}^{T^g}(t,s))^2 \tag{B4}$$

$$E^{climate}(p) = E^{CO_2}(p)E^{T^g}(p), \tag{B5}$$

where $c^{CO_2}$ is the concentration of $CO_2$, and $T_g$ is the global mean temperature. The parameter configuration with the lowest cost function in the $10^6$ member ensemble is used as an initial condition for an fmincon optimization, which adjusts the parameters to minimize $E^{climate}(p)$. Figure B1 shows the calibrated behavior of MiCES for reproducing the concentrations and temperature response of the CESM1-CAM5/MAGICC hybrid used to create the CMIP5 simulations.

5 **Appendix C:  Emissions scenario design**

The emissions scenarios developed for this study are not produced in an integrated assessment model. Rather, we use the strategy of Sanderson et al. (2016), which produced idealized emissions scenarios which follow RCP8.5 until a mitigation phase begins at time $t_m$, after which emissions follow a smooth trajectory (the derivative of emissions is continuous at time $t_m$), peaking and then decaying asymptotically to an 'emissions floor' ($E_m^0$). A parameter $t_m^{50}$ determines how quickly mitigation

10 occurs, and sets the length of time after $t^m$ that emissions are 50 percent of the way to the asymptotic emissions floor.

In this study, we add a second 'rampdown' phase to allow a period of intensive negative emissions followed by a long term relaxation of effort to achieve a stable temperature level. The rampdown phase follows a simple exponential decay, beginning at time $t_r$, with a long term asymptotic emissions level of $E_r^0$ and a relaxation time $t_r^{50}$, such that $CO_2$ emissions at time $t$ can be described as follows:

$$E(t) = \begin{cases} E_{RCP8.5}, & \text{if } t \leq t_m \\ A_m[(t - t_m^e)e^{-t/\tau_m}] - E_m^0, & \text{if } t_m < t \leq t_r \\ A_r[e^{-(t)/\tau_r}] - E_r^0, & \text{if } t > t_r \end{cases}$$

with 5 parameters to solve: $A_m$ and $A_r$, $t_m^e$, $\tau_m$ and $\tau_r$. These parameters can be solved using boundary conditions already established. We first solve for the parameters for the mitigation stage by fixing the emissions and rate of change of emissions at time $t_m$, as well as the timescale of mitigation:

$$E(t_m) = E_{RCP8.5}(t_m) \tag{C1}$$
$$\frac{dE}{dt}(t_m) = \frac{dE_{RCP8.5}}{dt}(t_m) \tag{C2}$$
$$E(t_m + t_m^{50}) = (E_{RCP8.5}(t_m) + E_0)/2. \tag{C3}$$

We can then solve for the rampdown parameters in a similar fashion by fixing the emissions at time $t_r$ to allow a smooth transition into the rampdown phase.

The parameters ($E_m^0$, $t_m^{50}$, $t_r$, $E_r^0$ and $t_r^{50}$) for the three scenarios were adjusted manually to achieve the desired temperature

25 behaviour: a stable 2°C climate, a stable 1.5°C climate and an overshoot of 1.5°C, with a stable 1.5°C climate post-2100. The parameters which were found to achieve these characteristics in the calibrated MiCES model are listed in table C1.

| Scenario | Year to begin mitigation phase | Time to 50% of long term (years, mitigation phase) | Mitigation phase emissions floor (PgC/yr) | Year to begin rampdown phase | Time to 50% of long term (years, rampdown phase) | Long term emissions level (PgC/yr, rampdown phase) |
|---|---|---|---|---|---|---|
| Parameter | $t_m$ | $t_m^{50}$ | $E_m^0$ | $t_r$ | $t_r^{50}$ | $E_r^0$ |
| 1.5 never-exceed | 2018 | 10 | -1.8 | 2065 | 35 | -0.3 |
| 2.0 never-exceed | 2018 | 25 | -0.85 | 2120 | 60 | -0.4 |
| 1.5 overshoot | 2018 | 18 | -4 | 2080 | 15 | -0.5 |

**Table C1.** Parameters for the three scenarios described in this study, using the functional form established in Sanderson et al. (2016), $t^m$ is the year that emissions depart from the RCP8.5 scenario and begin the mitigation phase, $t_m^{50}$ describes the rate of decarbonization in the mitigation phase, defined as the number of years after the mitigation phase start time that the emissions are equidistant between 2018 emissions and the emissions floor level, $E_m^0$ is the asymptotic level to which carbon emissions decay in the mitigation phase, $t_r$ is the year in which the mitigation phase ends and the rampdown phase begins, $t_r^{50}$ is the number of years after the rampdown start time that the emissions are equidistant between emissions in year $E_r^0$ and the long term emissions floor level and $e_f^r$ is the asymptotic level to which carbon emissions decay in the rampdown phase.

As in Sanderson et al. (2016), non-CO$_2$ greenhouse gas emissions depart from RCP8.5 at time $y_m$, and then decay asymptotically in a simple exponential to the RCP2.6 pathway using a timescale defined by $t^m$. Each follows RCP8.5 until $t^m$, and then decays to the respective RCP2.6 pathway with the decay constant $\tau_m$, such that for a gas $j$:

$$E^j(t) = \begin{cases} E_{RCP8.5}^j, & \text{if } t \leq t_m \\ E_{RCP8.5}^j e^{-(t-t_m)/\tau_m} + E_{RCP2.6}^j(1 - e^{-(t-t_m)/\tau_m}), & \text{otherwise} \end{cases}$$

5     Where $j$ can correspond to CH$_4$, N$_2$O, CO, VOCs, NOx, CFC-11, CFC-12, HFC-134a, HCFC-22 and SOx. There is no separate rampdown phase for non-CO2 emissions. Non greenhouse gas concentrations (aerosols, ozone etc.) follow RCP8.5 throughout the simulation.

*Author contributions.* Benjamin Sanderson designed the study and led the analysis and writing, Yangyang Xu and Jean-Francois Lamarque performed CESM simulations, Claudia Tebaldi and Michael Wehner performed extreme temperature analysis, Brian O'Neill provided input
10    on mitigation literature and conceptual planning, Alexandra Jahn performed sea-ice analysis, Flavio Lehner performed heat records analysis, Angeline Pendergrass participated in precipitation analysis, Warren G. Strand performed data post-processing, Lei Lin performed aridity analysis, Reto Knutti provided conceptual guidance and Jean-Francois Lamarque provided input on the simple model chemistry component.

*Acknowledgements.*

*Support.* This research was supported by the Regional and Global Climate Modeling Program (RGCM) of the U.S. Department of Energy's, Office of Science (BER), Cooperative Agreement DE-FC02-97ER62402.

*Correspondence.* Correspondence and requests for materials should be addressed to Benjamin Sanderson. (email: bsander@ucar.edu).

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

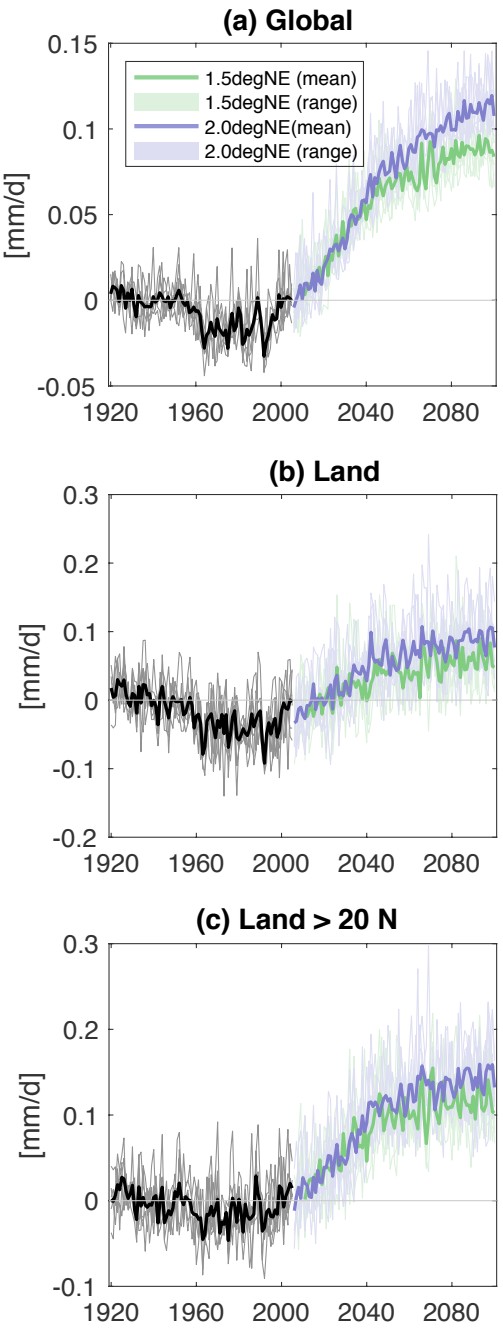

**Figure 6.** Changes in annual mean precipitation at the (a) Global, (b) Land-only and (c) high Northern latitude land. Values are relative to the 1921-1960 average. Grey lines show members of the historical CESM ensemble, while black line shows the historical mean. Thin colored lines show individual ensemble members for future scenarios, thick bold lines show the ensemble mean.

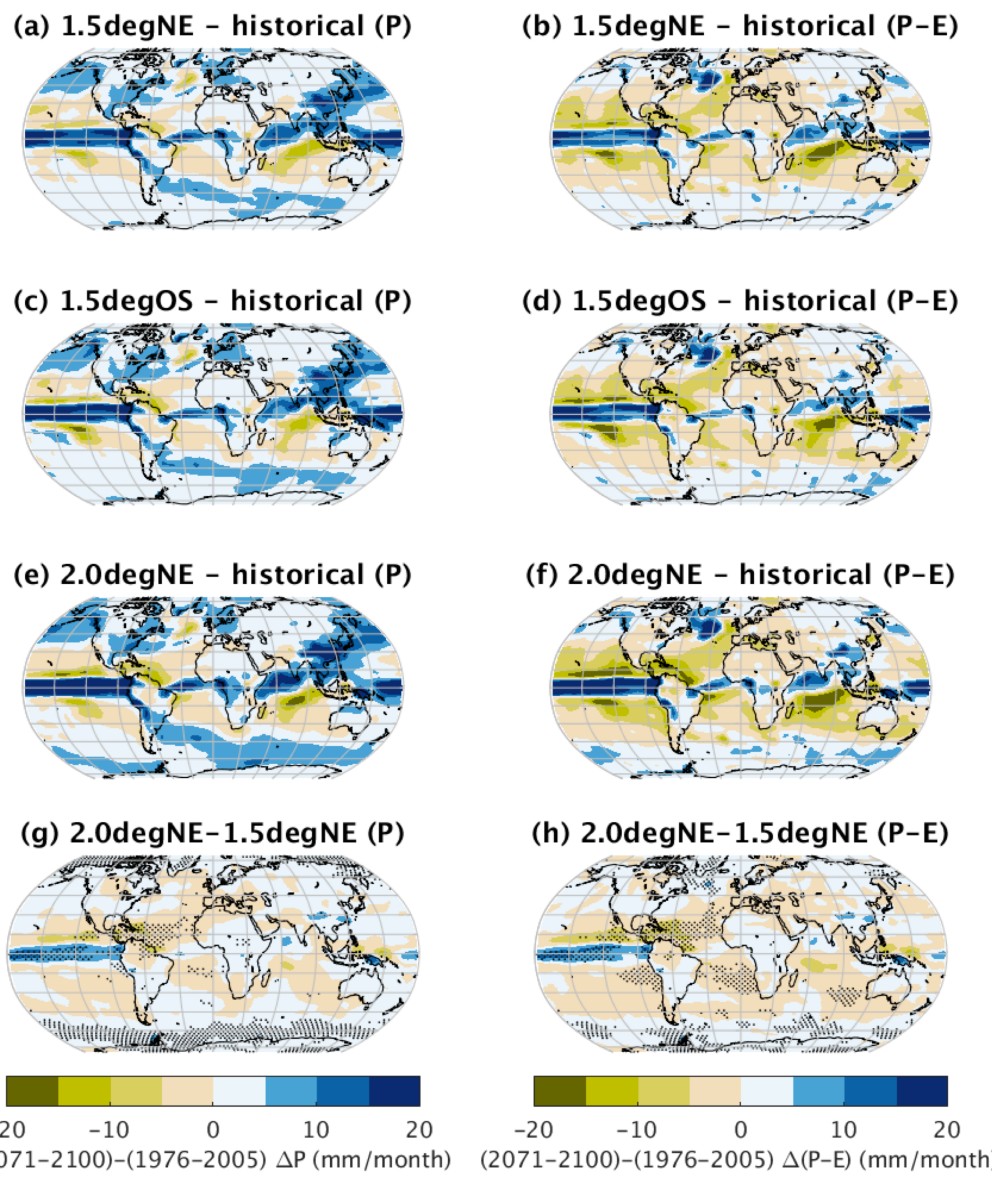

**Figure 7.** Maps showing ensemble mean 2071-2100 Precipitation and Precipitation-Evaporation changes from 1976-2005 historical conditions in *1.5degNE* (a,b), *1.5degOS* (c,d), *2.0degNE* (e,f) scenarios. Subplots (g,h) show the difference between mean 2080-2100 conditions in *1.5degNE* and *2.0degNE*, where significant regions are stippled. Significance is defined as a pixel in which the difference between the mean of the *2.0degNE* and *1.5degNE* ensembles exceeds the standard deviation of 2080-2100 values in the *2.0degNE* ensemble.

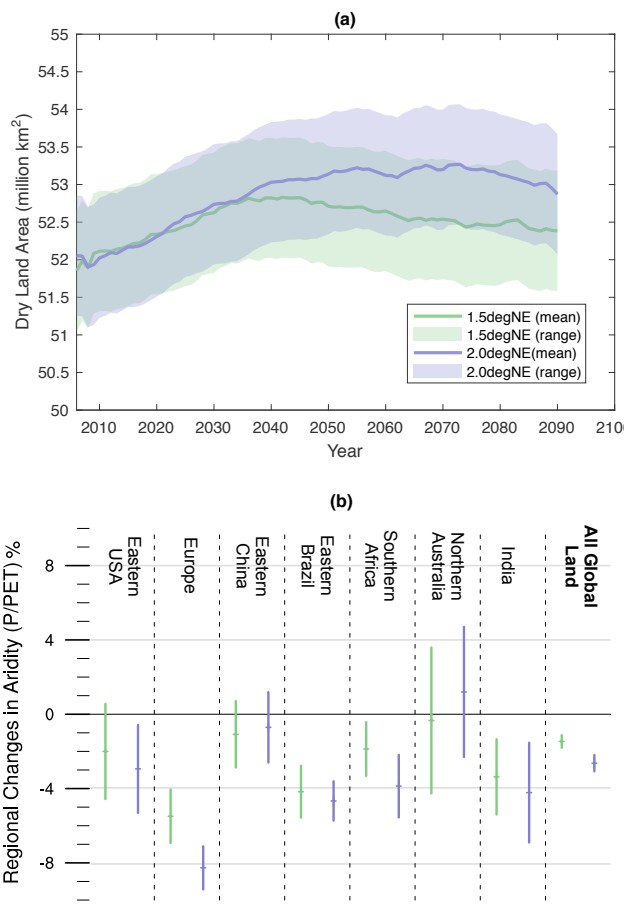

**Figure 8.** (a) Solid lines show the ensemble mean of aggregated land which qualifies as 'dry' land, where aridity (P/PET)<0.65. Lines are smoothed to show the 20 year running average value. Shaded areas are calculated as ±1 annual standard deviation from the mean in the respective ensemble. (b) shows the change in aridity for a number of specific regions in 2071-2100 relative to 1976-2005. Central dash indicates the ensemble mean value, while the vertical lines indicate the full spread in the *1.5degNE* and *2.0degNE* ensembles.

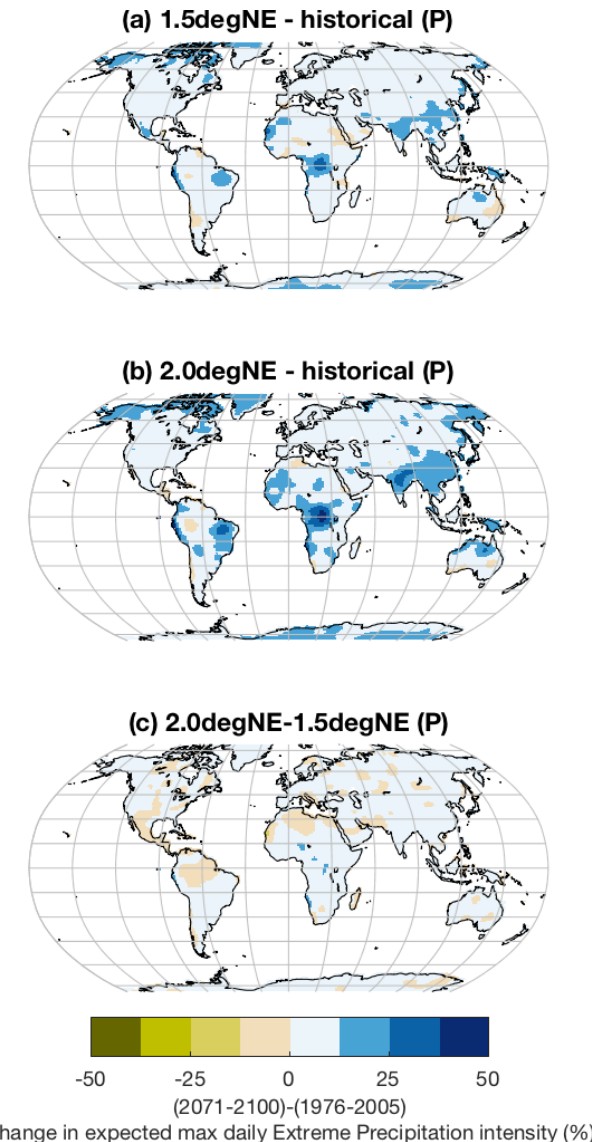

**Figure 9.** Maps showing the simulated percentage change in ensemble average annual maximum one-day precipitation for historical periods (1976-2005) and future (2071-2100) for (a) *1.5degNE*, (b) *2.0degNE* and (c) the difference between the scenarios. Maps are smoothed with a 2-D Gaussian smoothing kernel with standard deviation of 2 gridcells.

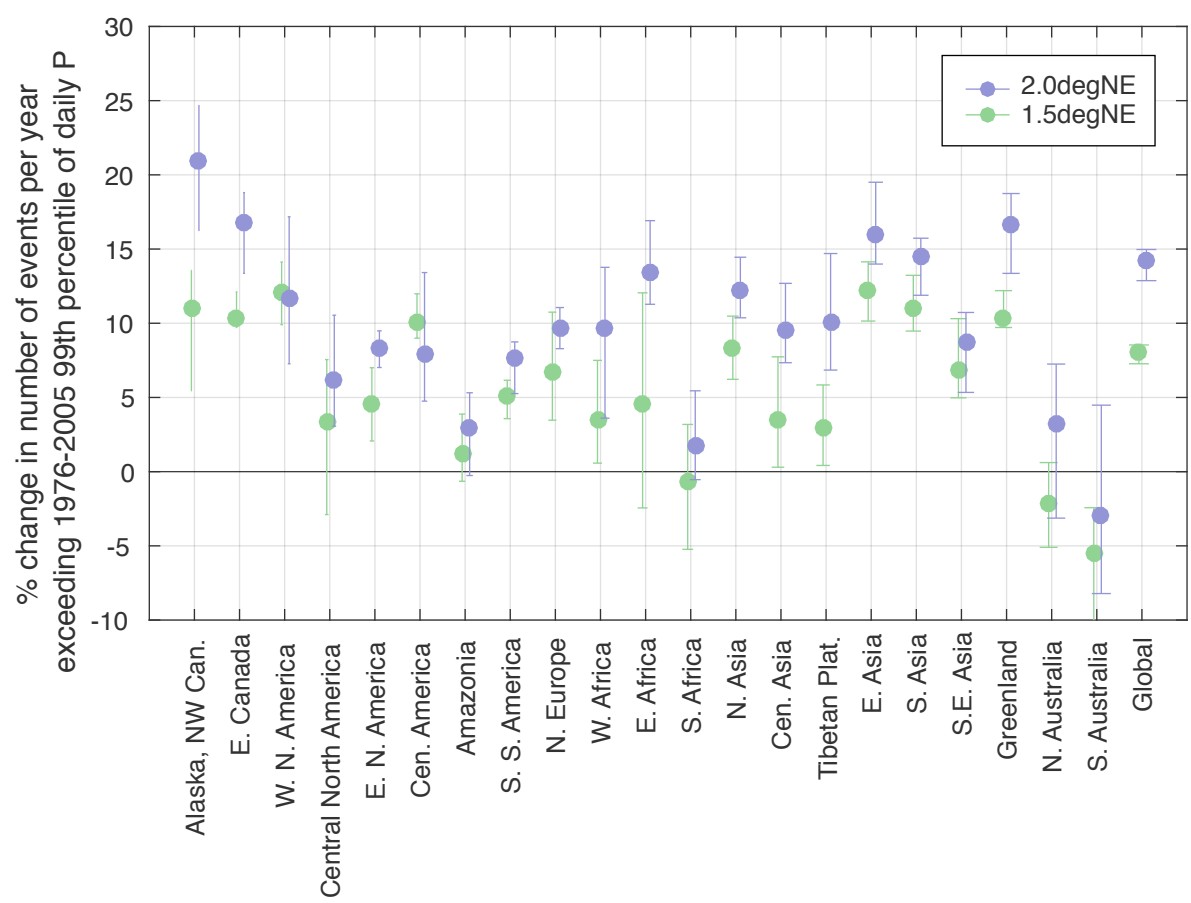

**Figure 10.** Aggregated percentage change in the aggregated frequency of exceedance of the historical 99th percentile of precipitation over regions as defined in Giorgi and Mearns (2002). Historical 99th percentile is calculated at a grid-cell level from the distribution daily precipitation from years 1976-2005, for ensemble members 1-10 (making 300 years of data). Events exceeding their point-level 99th percentile are summed over the relevant region for each ensemble member, such that the ensemble mean value of the number of exceedances represents the baseline. Future values are calculated from years 2071-2100 of *2.0degNE* and *1.5degNE*, again summing events which exceed the 1976-2005 99th percentile. The ensemble mean and range of percentage change in number of events in 2071-2100 relative to 1976-2005 is plotted.

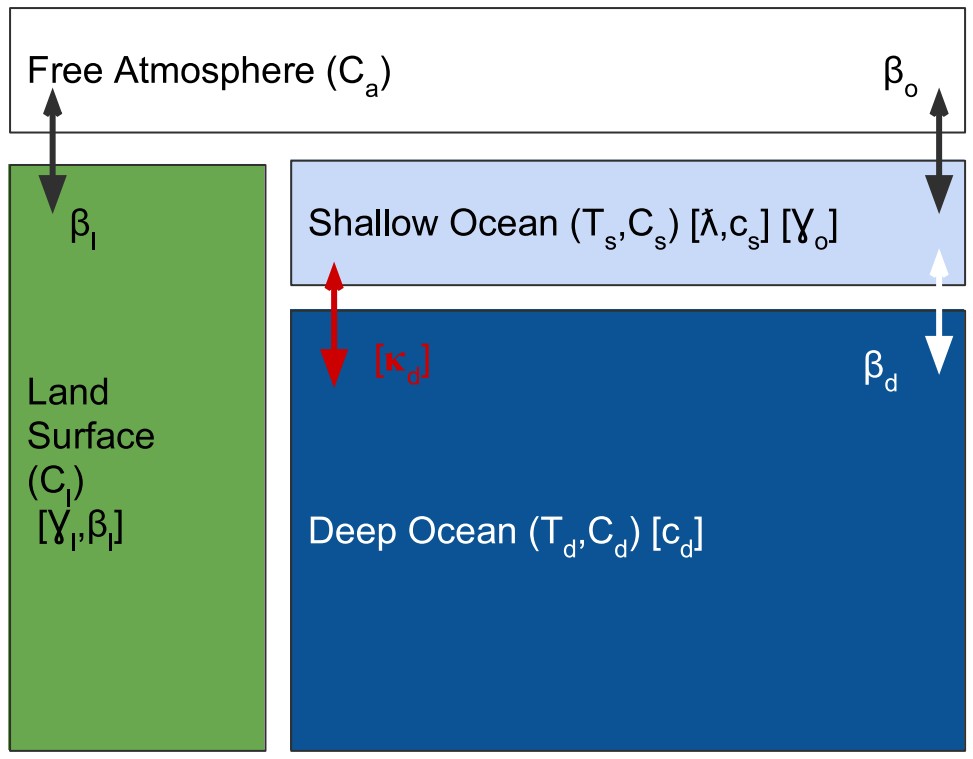

**Figure A1.** An illustration of the logic flow in the MiCES simple model

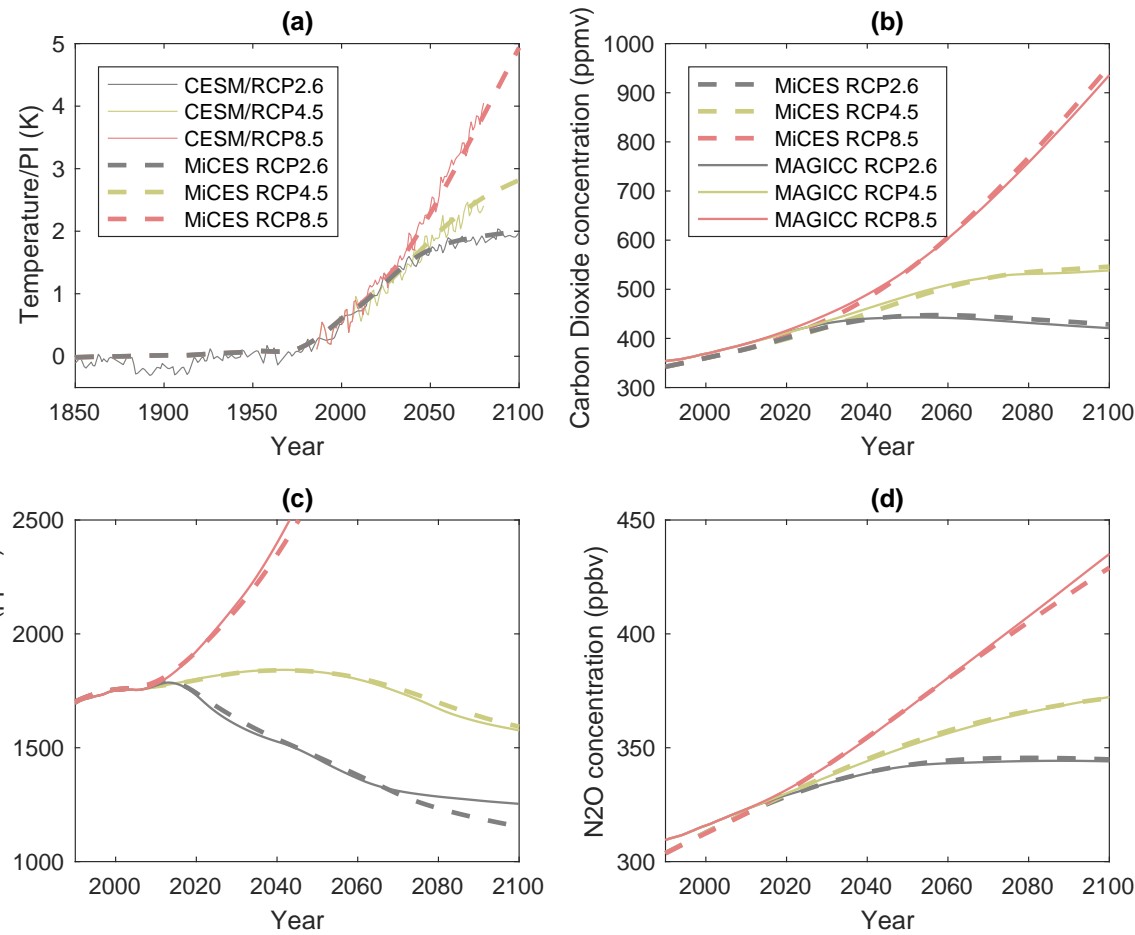

**Figure B1.** Calibrated behaviour of the MiCES model in the context of the temperature targets (subplot a, target is CESM1-CAM5) and the concentration targets (subplots b-d, where target is MAGICC). In each case, the target is represented by a solid line, and the optimized MiCES simulation is represented by a dashed line. The colors represent the three scenarios used in calibration.

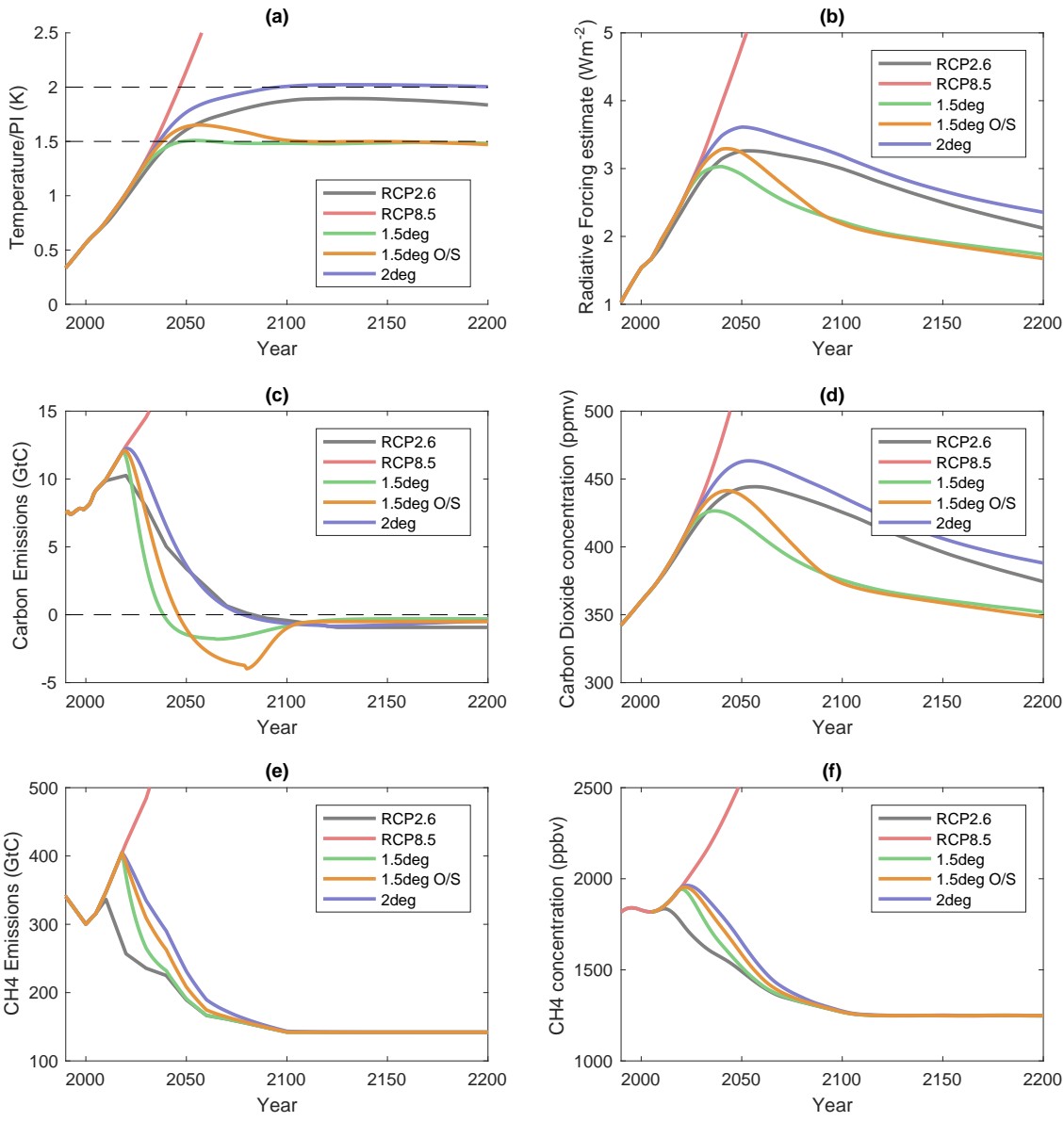

**Figure C1.** New scenarios used for this study. Figure (a) shows the predicted global mean temperature evolution, (b) shows the radiative forcing evolutions, (c) shows net anthropogenic carbon emissions (fossil fuel and land use combined), (d) shows $CO_2$ concentrations, (e) shows methane emissions and (f) shows the methane concentrations. Blue, green and orange lines are $2°C$, $1.5°C$ and $1.5°C$ overshoot respectively.

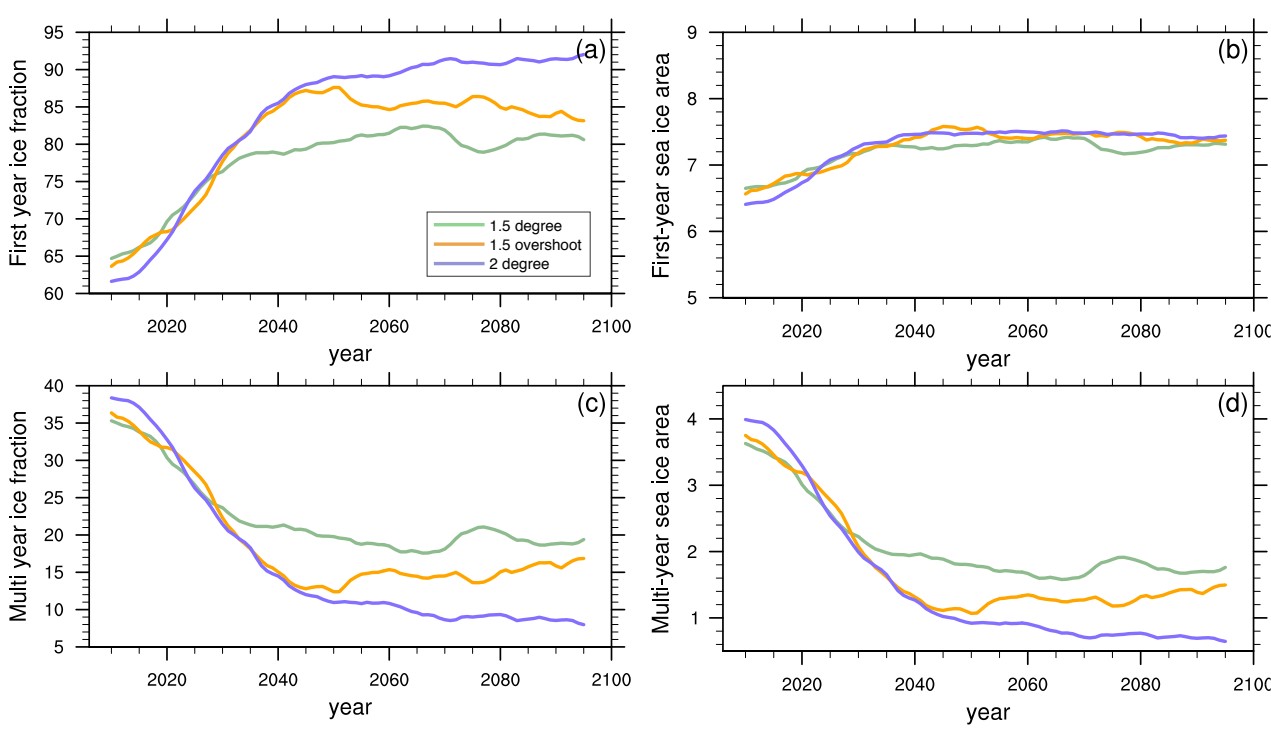

**Figure C2.** Evolution of the annual 10-year running ensemble-mean of first year and multi-year Arctic sea ice in the low emission simulations. (a) shows the fraction of Arctic sea ice which is less than a year old, (b) shows the sea ice area which is less than a year old, (c) shows the fraction of sea ice which is greater than a year old and (d) shows the total sea ice area which is greater than a year old.