# Peer review of "Community Climate Simulations to assess avoided impacts in 1.5°C and 2°C futures"

_Earth System Dynamics, 2017_

## Referee Comment (RC1) · D. Mitchell (Referee) · 15 May 2017

Summary

The authors document a new set of experiments specifically designed to address some of the Paris Agreement aims. The experiments are informed by use of an emulator, which allows for the specification of a set of scenarios to achieve 1.5 and 2 degrees for the CAM5 model. The authors document some of the impact-relevant changes in climate. Overall I think this is a well-written, very valuable addition to the literature surrounding Paris. The new set of simulations provides the community with an excellent data set to analyse, which also provides a complimentary method to the HAPPI approach.

I have many comments that should be addressed, but they are only small, so I would

recommend minor corrections. Three broad comments that the authors should consider are 1) to be a bit more upfront about the limitations of the CAM5 model, by, for instance including a paragraph on how well the model reproduces the past quantities that are looked at in the paper. Referring to AR5 etc, rather than new analysis could do this. 2) the nature of using a coupled ocean means that the climate may not be stable within the century. I think this needs to be clearer in the abstract, but also in discussion of Figure 1. 3) In the abstract, and throughout the text, it should be made clearer that this study is only a broad sweep of impact relevant analysis. For instance, each one of the sections on impacts could (and should) be multiple papers of analysis.

Signed Dann Mitchell

Comments

P1 L6: "impact-relevant long term climate data" – somewhere in the manuscript I think a sentence is needed about what variables are output. Is it everything from AR6? At this stage it is not clear what impact-relevant variables are stored.

P1 L11: "and only 1 in 40..." I think 'only' can be deleted here.

P1 L19 - P2 L1: "worlds emissions have been closer..." is this true for GHGs and aerosol etc? The interplay of both can be very different in different RCPs.

P2 L8-10: This sentence is not clear to me, I think the multi-model mean of CMIP5 RCP2.6 gives ∼1.5C, which seems in contradiction to your sentence. Of course it depends on pre-industrial definition.

P2 L12: "no individual model" should be "no individual fully coupled model".

P2 L18-26: I feel James et al, 2017, WIRES should be cited here, as it gives an overview of all these methods.

P2 L21: Can you make a comment on the time scales where the pattern scaling is relevant?

P2 L27: Acronym is wrong. It should read: "Prognosis and Projected Impacts". I think Mitchell et al, 2017b should also be added, which was the concept of HAPPI.

P2 L29: Please change 'will use' for 'uses', as these have already been performed.

P2 L30: "is computationally cheap", I think "so allows for huge ensembles to be run providing samples of extremes" or something similar should be added, for context.

P2 L34: "have the same SSTs..." actually there are tier 2 experiments which sample a much wider range of SST patterns (22 different patterns for 10 different years, which is 220 different patterns). Although I agree that is still doesn't sample the full range, as the CAM5 setup does.

P4 L5: I think this should be "Figure 1(a) and 1(b)". Otherwise 1(a) is not referenced. But it makes sense to refer to both panels here anyway.

P4 L11: "1850-1920", can you provide some justification for choosing this period please.

P5 L13: I think "until 2100" can be removed, otherwise it sounds like it changes there (but actually just the simulations stop there).

P5 L14-15: I do not think the plots look as stable as the authors say, especially not for the 2C experiment. And I wouldn't see how they would be with the long term ocean time scales. Can the authors expand on this section.

P5 L16: "large scale climate" sounds unclear. I think "global-scale climate" is better.

P5 L20-25: I would note that within uncertainty there is no change over the scenarios.

P7: I think some more text on the reproducibility of the Arctic response is needed, especially in the historical period. There is a large range in CMIP5 projections.

Figure 2: It would be nice to see a bias plot from the model (historical versus CRU, or something similar). This might address the comment above to some extent.

P8 L12: I think a stronger reference for this point is Gasparrini et al, 2016. You could either add it on, or replace the old reference with it. Also it would be best to put an 'e.g.' in front of these.

P8 L28-end: I didn't follow the use of the GEV distribution here. You have a large sample size, 10 ensemble members of 30 years), so you should be able to calculate a return period from that? Why model it? Perhaps it is to use a covariate and account for the transient response? I think this needs more clarification.

P12 L13-17: It is not clear to me if land use/cover is different in the different scenarios. This will be important here, so should be clarified.

Conclusions/discussions: I think this section should be shorter. I like the discussion part but feel other results are repeated in too much detail.

P13 L17: "simulations to our knowledge" please change to "simulations using a fully coupled model to our knowledge".

P13 L32: "longer than a century" please add "and therefore were not assessed in this study".

P14 L20-27: I feel this section could be a bit more balanced between this study and HAPPI. I.e. there are advantages and disadvantages to both. Also, note that 2 of the HAPPI models have done sister experiments which include some form of ocean coupling. i.e. the NorESM2 models, and the MetUM-GOLM model. In both cases, they use slab oceans.

References

Gasparrini, Antonio, et al. "Mortality risk attributable to high and low ambient temperature: a multicountry observational study." The Lancet 386.9991 (2015): 369-375.

James, Rachel, et al. "Characterising half a degree difference: a review of methods for identifying regional climate responses to global warming targets." Wiley Interdisci-

plinary Reviews: Climate Change (2017).

Mitchell, Daniel, et al. "Realizing the impacts of a 1.5 [deg] C warmer world." Nature Climate Change (2016).
* * *

---

## Referee Comment (RC2) · M. Sarofim (Referee) · 19 May 2017

Sanderson et al. have compiled a very useful set of climate simulations to aid the community in comparing a 1.5 degree future to a 2 degree future. This paper does a good job of laying out the methodology, as well as presenting a limited set of analyses of interesting impacts. The writing is clear, and the figures are informative and well-presented. I also commend the authors on providing links to open-source code for the simple model as well as the data for the simulations.

I do have a number of minor comments for the authors to consider when revising the manuscript for publication. Two in particular might require some recalculations (see line-by-line comments below for more detail). First, I would recommend using the more recent Kopp et al. semi-empirical equations in place of the Rahmstorf et al. (2007)

version. Second, I would consider comparing future model results against historical records from the model, rather than from observations, in Figure 3 and related text. I can understand not taking the latter recommendation, but in that case, I would include text discussing the how differences between model and observations during the 1976-2005 historical period might impact the results. Other than that, most of my comments below are generally minor sentence edits.

Line by line comments:

Page 1, Line 17: I would also reference Fawcett et al. here as it has a good analysis of the probability of staying below various thresholds for the RCPs (based on simple models, not full GCMs, of course) (DOI: 10.1126/science.aad5761)

Page 2, Line 5: I am unclear how the authors determined that this commitment would lie "on the verge of economic and physical plausibility": as I read Smith et al., they evaluate the cost & capacity of negative emissions technologies, but do not define a plausible upper bound to how fast society could implement these technologies if properly motivated. I would rephrase this sentence to be more judicious, e.g., "likely requiring substantial commitment to negative net carbon emission technologies in the 2nd half of the century".

Page 2, Line 19-20 and 23-24: it seems to me that "finding time periods from other scenarios" is equivalent to "another approach is to 'time-shift' by taking periods...": I would delete one of these descriptions.

Page 4, line 14: delete "in a decade" (it is redundant with "by 2027").

Page 5, line 2-3: I understand why the authors made the justifiable choice of using RCP8.5 non-GHG forcings for all the scenarios – however, it would be useful to have a brief aside that notes that a 1.5 degree scenario that is internally self-consistent might look slightly different than a 1.5 degree scenario that is a hybrid between RCP8.5 non-GHG forcings and low-GHG-concentrations. In particular, sulfur emissions might be

even lower in a 1.5 degree scenario (like RCP2.6 has lower sulfur than RCP8.5, though comparing RCPs should also be done with caution because they come from different IAMs so not all differences are necessarily due to policy effects), which would require even tighter GHG emissions reductions, but more relevantly for the paper, might also impact temperature patterns because aerosols have different land/ocean and hemispheric forcings than well-mixed GHGs. Similarly, a 1.5 degree future that relies heavily on bio-fuels would have very different land-albedo forcings. I also note that the choice of setting RCP2.6 as a limit to non-CO2 GHG reductions has an influence as well, possibly contributing to the necessity for the long-term CO2 emission floor to be negative (Table C1, column 7) because the CFCs and maybe N2O have lifetimes long enough that their concentrations would not have stabilized by 2200, requiring ongoing offsetting negative CO2 emissions (I think this study does not include SF6 or PFCs).

i would also potentially be curious regarding where CESM falls in terms of climate sensitivity in the larger CMIP universe, which would also determine how aggressive the GHG mitigation would need to be to stay below given targets.

Page 5, line 3: Fix parentheses – in Kay et al. (2015).

Page 5, line 18-20: I would suggest replacing the Rahmstorf (2007) semi-empirical approach with that from Kopp et al. (2016) (doi: 10.1073/pnas.1517056113): Rahmstorf is a co-author of the latter paper, which claims to reconcile the semi-empirical approach with process-based models, and therefore I would consider this to be an improved update to the Rahmstorf equations. (a correction would still be required for ice sheet melt, so Horton et al. 2014 might still be appropriate there).

Page 8, Line 15-16 (and Figure 3 generally): I note that during the period 1976-2005, models have already exceeded the maximum historical observed temperatures from 1976-2005. This makes me wonder whether the appropriate comparison should be between future model and historical model, rather than future model compared to historical observations. Either that, or there should be a discussion of this potential dis-

СЗ

crepancy. Reading Lehner et al., I think that paper did do compare to historical models – e.g., Figure 3 has both the observed 1920-2014 and the model 1920-2014, so that one can compare model-future to model-past, e.g., like-to-like. I recognize that there is still the potential for model bias to creep in here (as discussed in Lehner): if the model has more variability than the observed, then it is harder to exceed records, and vice versa, but I think a like-to-like is a cleaner comparison.

In addition, for section 3.3, I don't see a cited source for the observed temperatures – is it BEST as in Lehner? Or ERAi?

I would suggest an additional paragraph here which could do several things: 1) note the source of observed temperature data, 2) discuss how well the model reproduces the observations over the 1980-2005 period (and/or if any bias-correction is being used here), 3) discuss the model-observation comparison over the 2006-2016 period.

Page 8, Line 26-27: Related, is this "noteworthy" statement regarding the 2006-2016 period? Please clarify, and see above suggestion.

pg 8, line 33-34: 1) some regions experience a greater increase in extreme than in mean: is the opposite true as well? 2) following up on that: it would be very interesting to have a quantitative estimate of this effect: e.g., x% of the land area experiences a warming of extremes more than 50% faster than the mean, while y% experiences a warming 50% slower than the mean. Or, averaged across land areas, extremes warm x% faster than the mean. Or something like that.

pg 9, line 9: I might note that the greater signal to noise is seen at lower latitudes even though absolute warming at those latitudes is generally smaller (which has the opposite effect of there being less variability at those latitudes).

pg. 21, line 5: stray period should be deleted.

Figure 2: "subplot" should be singular.

Figure 4: Is the historical period 1976-2005? Please specify. Also, it is based on

observed (like Figure 3) or on modeled historical (like I think that most other figures do)

Figure 6: legend needs more detail: I assume that black is modeled historical, but it could potentially be observed. Also, what's the time period of smoothing – annual?

Figure C2: Please include a legend for the colors as in Figure 1. Also, I don't think it that this figure extends far enough to demonstrate this, but I'd be curious about whether the 1.5NE and the 1.5OS could be used to investigate path dependence/memory/inertia. For example, one might expect some additional warming of the Arctic Ocean during the years in which the temperature is above 1.5 degrees which might take a number of years to dissipate even after the global surface air temperatures have returned to 1.5 degrees, which might lead to slightly lower sea ice extent in the 1.5OS case than then 1.5NE case for some years after stabilizing back at 1.5 degrees.

(One could go further, and imagine hypothetical tipping points that could be exceeded in the 1.5 OS case which would not be resolved by cooling back down to 1.5 degrees, but I would imagine that this would be somewhat unlikely, and even if such a tipping point existed, this modeling system might not be able to catch it).

(sea level could be a place where there might be some long-term memory of a brief excursion to 1.7 degrees, as in Zickfeld et al., doi: 10.1073/pnas.1612066114: Figure 1(c) seems to show this – it might be interesting to note the divergence between 1.5OS and 1.5NE at 2100 and what date the two scenarios become equal, if they ever do)

Table B1: Would it be possible to include an additional column with the values of each parameter that resulted from the calibration process?

---

## Referee Comment (RC3) · KT Tanaka (Referee) · 31 May 2017

The paper provides a first assessment of impact-relevant climate change at the 1.5 and 2°C warming levels based on an earth system model CESM. The authors describe and discuss the results of model simulations specifically designed to analyze these two temperature goals in the stabilization context of the Paris Agreement. They present various facets of climate change under 1.5 and 2°C stabilizations, including mean temperature, extreme temperature, mean precipitation, extreme precipitation, sea level rise, and sea ice. Among several significant differences between the 1.5 and 2°C cases that are identified, the most drastic is the probability of September ice-free Arctic. Furthermore, they develop and apply a simple climate model to calculate inversely emissions scenarios that lead to desired temperature stabilization goals. Obviously, lots of efforts have been put into this paper. The paper is very clearly written and the results are also

clearly presented. I think it would be one of the key papers informing the debates on the 1.5 and 2°C targets. I have several minor comments as laid out below. If these are sufficiently addressed, I would formally support publication of this paper in Earth System Dynamics.

1) I start with a broad comment related to the interpretation of the results. The paper ends with the statement stressing the differences in impacts between 1.5°C to the 2°C levels: "Irrespective of feasibility, these simulations indicate that a relaxation of ambition from the 1.5°C to the 2°C level would result in significantly greater impacts at the global scale, in the tropics and at high latitudes." The abstract also highlights the differences, rather than the similarities: "Exceedance of historical record temperature occurs with 60 percent greater frequency in the 2°C climate than in a 1.5°C climate aggregated globally, and with twice the frequency in equatorial and arid regions. Extreme precipitation intensity is statistically significantly higher in a 2.0°C climate than a 1.5°C climate in several regions. The model exhibits large differences in the Arctic which is ice-free with a frequency of 1 in 3 years in the 2.0°C scenario, and only 1 in 40 years in the 1.5°C scenario." I take issue with the direction of argument, which is somewhat implicit in this paper. The paper makes me wonder what are the motivations. It is perhaps too broad to raise this here, but given the upcoming IPCC Special Report on Global Warming of 1.5°C, are we as a community in charge of concluding urgently that there are discernable differences in impacts between 1.5 and 2°C warming levels? The reason why I am raising this is that my overall impression of the results is drawn more toward the similarities. Visual inspection of the series of results certainly shows that there are significant (but not drastic, except for the sea ice (Fig. 1)) differences for various metrics (e.g. extreme precipitation (Fig. 10)) at the global mean level. But when it comes to regional and grid levels, differences are generally obscured by spatial and temporal variability as indicated by overlapping uncertainty ranges (just like any other global climate projections). In other words, similarities are more dominant than differences in my eyes. If there were multiple models performing the simulations, regional differences could be even less tantalizing. As a suggestion, I would think it is

worth pointing out the similarities, not just the differences, at the abstract level. If the authors wish to bring forward only the differences, I would suggest that the basis of judgement be clarified to substantiate the claim.

2) In my view, comparisons between 1.5degNE and 1.5degOS results are worthy of more discussion especially in the final section of the paper because it informs what the overshoot means in the context of 1.5°C stabilization. It is unclear how the Paris Agreement would deal with an overshoot from the Agreement text. But, given the closing door for the 1.5°C target as pointed out in this paper (page 15, line 5), possibilities of overshooting the target before achieving it are ever more relevant. As far as I am aware, implications of overshoot in the context of 1.5°C target are not specifically analyzed in previous studies (e.g. (Rogelj et al. 2015)). I think a more dedicated discussion on the comparison between 1.5degNE and 1.5degOS results would thus be useful.

3) Fig 1 shows that significantly negative CO2 emissions (about -2 GtC/yr in average) for more than 50 years (1.5degNE case) do not lead to a decline in the global-mean temperature. It is a removal of roughly 100 GtC from the atmosphere. I think this appears at odd with the rule of thumb that the stabilization level is determined by the cumulative CO2 emissions (Allen et al. 2009). Is there any explanation or perhaps some references that help clarify this temperature response?

4) While the carbon in the land surface (as C sub I) is shown in Fig A1, it does not seem to be the case from the text that the land carbon cycle itself is explicitly modeled. Only the climate-land carbon cycle feedback is provided without being linked to the land carbon mass (Equation (A2)). Furthermore, in many simple climate models, CO2 fertilization effect is modeled as a logarithmic function of the fractional increase of atmospheric CO2 concentration from preindustrial level (e.g. see equation (2.1.50) in page 28 of (Tanaka et al. 2007)). On the other hand, Equation (A2) indicates that CO2 fertilization effect is not a function of atmospheric CO2 concentration. These points need to be clarified because applicable ranges of this model may be limited to low scenarios because of the treatment of carbon cycle-related feedbacks.

5) The paper says in page 3 "Our main design choice was to minimize the number of its degrees of freedom to allow for fast calibration to reproduce the global mean trajectory of any given GCM." But when I look at the number of parameters, especially those for CH4 and N2O, I must say it is not really a model of minimal complexity. As some of the co-authors are aware, I developed a simple climate model (Tanaka et al. 2007; Tanaka et al. 2009), which I consider simple but not minimal at all. Even my model has less tunable parameters for CH4 and N2O (Table 3.2 of (Tanaka et al. 2007)). But this is just a naming issue, not a scientific one. Nevertheless, I do not understand some of the parameters in Table B1. For instance, the present-day growth rates for CH4 and N2O (ppb/a) and the present-day concentration of N2O should be model outputs, rather than model parameters because it is stated in page 17 lines 10-11: "The inputs to MiCES are global total emissions of greenhouse gas emissions (CO2, CH4, N2O, CFCs, HCFCs, CO)." This requires a clarification.

Technical comments:

Appendix A The notation for the conversion factor between ocean carbon content in Pg and ocean carbon concentration is not consistent. It is rho in some places but rho sub o in other places.

Page 15: Line 25 The sentence is unfinished.

Page 15: Equation (A1) One of the brackets is not closed.

Page 16: Line 18 Perhaps "due to" instead of "due"?

References

Allen MR, Frame DJ, Huntingford C, Jones CD, Lowe JA, Meinshausen M, Meinshausen N (2009) Warming caused by cumulative carbon emissions towards the trillionth tonne. Nature 458 (7242):1163-1166. doi:10.1038/nature08019

Rogelj J, Luderer G, Pietzcker RC, Kriegler E, Schaeffer M, Krey V, Riahi K (2015) Energy system transformations for limiting end-of-century warming to below 1.5 [deg]C. Nature Clim Change 5 (6):519-527. doi:10.1038/nclimate2572

Tanaka K, Kriegler E, Bruckner T, Hooss G, Knorr W, Raddatz T (2007) Aggregated Carbon Cycle, Atmospheric Chemistry, and Climate Model (ACC2) – description of the forward and inverse modes. Reports on Earth System Science, vol 40. Max Planck Institute for Meteorology, Hamburg

Tanaka K, Raddatz T, O'Neill BC, Reick CH (2009) Insufficient forcing uncertainty underestimates the risk of high climate sensitivity. Geophys Res Lett 36 (16):L16709. doi:10.1029/2009gl039642

---

## Author Comment (AC4) · 29 Jun 2017

And of course, many thanks to the reviewer for their thoughtful reading of the paper and for their wider contributions to this topic!

---

## Author Response (AR1)

**Response to Reviewer 1 (D. Mitchell)**

*I have many comments that should be addressed, but they are only small, so I would recommend minor corrections. Three broad comments that the authors should consider are*

*1) to be a bit more upfront about the limitations of the CAM5 model, by, for instance including a paragraph on how well the model reproduces the past quantities that are looked at in the paper. Referring to AR5 etc, rather than new analysis could do this.*

Thanks.  We added the following paragraph to this point:
"CESM1-CAM5 is a single climate model, and like any model is subject to biases in both its present day simulations and in future projections.  As such, the ensemble spread in this study does not represent true uncertainty in future projections, rather a single estimate of climate evolution.  However, multi-model assessments in  the past have indicated that CESM1-CAM5 is one of the better performing models in the CMIP5 archive.  \cite{sanderson2015representative} found this model to be the best performing in a selection of mean state metrics, and \cite{flatocoauthors}, Figure 9.37 shows that the model has one of the better simulations of extreme temperature and precipitation metrics in the CMIP5 archive."

*2) the nature of using a coupled ocean means that the climate may not be stable within the century. I think this needs to be clearer in the abstract, but also in discussion of Figure 1.*

Added this to the abstract to make clear that the emulator was only tested in the 21st century

"These scenarios … **are realized (for the 21st century) in the coupled model** and are freely available to the community."

And added the following to the temperature evolution results:

**"Note that although these results suggest that the predicted emulation of stable temperatures in the emulator is validated for the evolution of global temperatures in the coupled system in the 21st century, there may be ocean dynamical processes at longer timescales in the coupled model which are not represented in the emulator's thermodynamic ocean.  As such, we have not tested the stability of global temperatures at multi-century timescales with these emissions pathways."**

*3) In the abstract, and throughout the text, it should be made clearer that this study is only a broad sweep of impact relevant analysis. For instance, each one of the sections on impacts could (and should) be multiple papers of analysis.*

Changed abstract as follows:
"Here we describe the design of the simulations and **a brief overview** of their impact-relevant climate response. "

Added this to the end of the introduction:

"We aim in this study to provide a short overview of differences in impact-relevant climate variables, with the hope that further studies will focus in more detail on specific processes, regions or societal impacts."

*Comments P1 L6: "impact-relevant long term climate data" – somewhere in the manuscript I think a sentence is needed about what variables are output. Is it everything from AR6? At this stage it is not clear what impact-relevant variables are stored.*

The list is far too long to print in the paper. We've added a link to an online table:
http://www.cesm.ucar.edu/projects/community-projects/LENS/data-sets.html

*P1 L11: "and only 1 in 40. . ." I think 'only' can be deleted here.*

done

*P1 L19 - P2 L1: "worlds emissions have been closer. . ." is this true for GHGs and aerosol etc? The interplay of both can be very different in different RCPs.*

Expanded as follows:
"Since then, the world's greenhouse gas emissions have been closer to the highest emissions pathway (RCP8.5) than any other, even accounting for a recent slowdown in emissions growth \citep{van2011representative,quere2016global} (aerosol evolution differs relatively much less between RCPs, which are so far broadly in line with observations \cite{klimont2013last})."

*P2 L8-10: This sentence is not clear to me, I think the multi-model mean of CMIP5 RCP2.6 gives ~1.5C, which seems in contradiction to your sentence. Of course it depends on pre-industrial definition.*

Agreed.  Changed as follows:

"Although some individual models exhibited less than 1.5C warming in this scenario (median warming was 1,6C), CESM warming in this scenario is closer to 2 degrees \citep{meehl2013climate}."

*P2 L12: "no individual model" should be "no individual fully coupled model".*
Done

*P2 L18-26: I feel James et al, 2017, WIRES should be cited here, as it gives an overview of all these methods.*
Done.

*P2 L21: Can you make a comment on the time scales where the pattern scaling is relevant?*

Not sure what is meant by this comment.

*P2 L27: Acronym is wrong. It should read: "Prognosis and Projected Impacts".*
Updated,

*I think Mitchell et al, 2017b should also be added, which was the concept of HAPPI.*

Added.

*P2 L29: Please change 'will use' for 'uses', as these have already been performed.*

Done.

*P2 L30: "is computationally cheap", I think "so allows for huge ensembles to be run providing samples of extremes" or something similar should be added, for context.*

Done.

*P2 L34: "have the same SSTs. . ." actually there are tier 2 experiments which sample a much wider range of SST patterns (22 different patterns for 10 different years, which is 220 different patterns). Although I agree that is still doesn't sample the full range, as the CAM5 setup does.*

Changed as follows:
"Because simulations in HAPPI will have one of a finite set of predefined SST patterns, the estimate of significance of the difference in climate states will not completely sample ocean-driven variability."

*P4 L5: I think this should be "Figure 1(a) and 1(b)". Otherwise 1(a) is not referenced. But it makes sense to refer to both panels here anyway.*

It was referenced in the previous section, but we've now included it here as well.

*P4 L11: "1850-1920", can you provide some justification for choosing this period please.*

The large ensemble initial conditions (which we used for these simulations) did not diverge before 1920 (i.e. there's only one ensemble member until then, after which they branch). As such, using 1850-1880 would have been rather noisy - hence we used the longer time period. We've added a sentence to this effect:
**"(where pre-industrial is taken as the 1850-1920 mean - averaging over the period before the model initial conditions were branched, and using a 70 year rather than 30 or 50 year mean because there is only one ensemble member for this period). "**

*P5 L13: I think "until 2100" can be removed, otherwise it sounds like it changes there (but actually just the simulations stop there).*

done

*P5 L14-15: I do not think the plots look as stable as the authors say, especially not for the 2C experiment. And I wouldn't see how they would be with the long term ocean time scales. Can the authors expand on this section.*

We've added the caveat section in response to your major point above, and we've re-worded the paragraph to remove the word "stabilized":

  The \emph{1.5degNE} scenario reaches 1.5C above pre-industrial levels and then maintains this temperature until 2100.  The \emph{1.5degOS} scenario reaches a peak temperature in 2050 of 1.7${^\circ}$ above pre-industrial before cooling to 1.5C by 2100, and the \emph{2.0degNE} scenario reaches slightly over 2.1C by 2100.

*P5 L16: "large scale climate" sounds unclear. I think "global-scale climate" is better.*

done

*P5 L20-25: I would note that within uncertainty there is no change over the scenarios.*

We already say " The inherent uncertainty in the sea level response at a given emissions level is greater than the difference between the scenarios considered here. "

*P7: I think some more text on the reproducibility of the Arctic response is needed, especially in the historical period. There is a large range in CMIP5 projections.*

Added: "And while it should be noted that there is large diversity in the rate of loss of Arctic sea ice in CMIP5 models \citep{mahlstein2012september}, the CESM sea-ice loss per degree of global warming to date is less than that which has been observed \citep{Rosenblum_Eisenman_2017}. "

*Figure 2: It would be nice to see a bias plot from the model (historical versus CRU, or something similar). This might address the comment above to some extent.*

There isn't space to evaluate CESM's historical performance here and I would argue that (1) it's already been comprehensively done and (2) the mean state temperature bias plot tells you relatively little (if anything) about the temperature response (otherwise we would be using it to constrain the response).

*P8 L12: I think a stronger reference for this point is Gasparrini et al, 2016. You could either add it on, or replace the old reference with it. Also it would be best to put an 'e.g.' in front of these.*

Added as suggested

*P8 L28-end: I didn't follow the use of the GEV distribution here. You have a large sample size, 10 ensemble members of 30 years), so you should be able to calculate a return period from that? Why model it? Perhaps it is to use a covariate and account for the transient response? I think this needs more clarification.*

The clarification is well explained in the source reference Tebaldi and Wehner (2016). At the grid-point level, the sample size is insufficient to provide a smooth empirical distribution for this level of extreme event, so the extreme value analysis is necessary, even with a 10 member ensemble.

*P12 L13-17: It is not clear to me if land use/cover is different in the different scenarios. This will be important here, so should be clarified.*

We already detail this in the introduction:
" In each of the low emission scenarios in this paper, only well-mixed greenhouse gas concentration are changed between scenarios, all other forcings (land use, aerosol emissions, and ozone) follow RCP8.5 throughout the 21$^{st}$ century as in \citep{kay2015community})."

*Conclusions/discussions: I think this section should be shorter. I like the discussion part but feel other results are repeated in too much detail.*

We have removed 2-3 paragraphs which repeat information from the results section.

*P13 L17: "simulations to our knowledge" please change to "simulations using a fully coupled model to our knowledge".*

Done

*P13 L32: "longer than a century" please add "and therefore were not assessed in this study".*

This paragraph is now removed

*P14 L20-27: I feel this section could be a bit more balanced between this study and HAPPI. I.e. there are advantages and disadvantages to both. Also, note that 2 of the HAPPI models have done sister experiments which include some form of ocean coupling. i.e. the NorESM2 models, and the MetUM-GOLM model. In both cases, they use slab oceans.*

Agreed.  Re-written as follows:
"A key factor determining whether or not there are significant differences between 1.5C and 2C of warming is the magnitude of the climate system response to a difference of half a degree of warming relative to internal climate variability. This question can be informed by ensembles of coupled climate model simulations including estimates of internal variability arising from the ocean and atmosphere, and yet.  These experiments should hopefully provide context (for a single model) for the primary multi-model effort being pursued by the community to address the difference between 1.5C and 2C of warming (HAPPI, \cite{mitchell2017half}) which relies on AMIP simulations with prescribed SSTs."

**Response to reviewer 2 (Marcus Sarofim)**

First, I would recommend using the more recent Kopp et al. semi-empirical equations in place of the Rahmstorf et al. (2007) C1 version.

*Done - using the 10th and 90th percentiles of Kopp's distributions of the 'a' and 'c' parameters to define the uncertainty range.*

Second, I would consider comparing future model results against historical records from the model, rather than from observations, in Figure 3 and related text. I can understand not taking the latter recommendation, but in that case, I would include text discussing the how differences between model and observations during the 1976- 2005 historical period might impact the results.

*Thanks for the suggestion, but for a number of reasons we would like to keep the figure as a direct comparison to observations. However, the figure produced in the way you suggest looks as follows:*

[Figure]

[Figure]

*The results are very similar as in the original Figure 3, except for a mean offset that comes from the fact that you look for a record in 10 ensemble members rather than 1 observational record.  As such, any given record will very likely be higher because it is based on 10x more samples, making that record harder to break in the future. One could pick a record from one arbitrarily individual simulation, but we feel that would make the figure less relevant as a communication tool.  Using all ensemble members, all the time series in Fig 3 would shift down but basically keep their shape and relative position. This would imply about ~30% less land fraction exposed when you look for the record in the 10 ensemble members. As such, and for consistency with Lehner (2016), we would prefer to keep the plot as-is.*

*We added the following caveat:*

*"It is notable that some records in Figure \ref{record} are exceeded before 2005 because the historical evolution of the CESM ensemble differs from the real-world historical evolution and there could potentially be some model regional model biases. However, the behaviour of CESM in the period 2006-2016 is in within the range of model record exceedance (both globally and in each of the regions considered), giving confidence that regional biases are not strongly influencing this metric.  Note that if the records were taken from the historical simulations of CESM itself for consistency, almost all historical records would be a higher temperature because the effective sample period in a 10 member ensemble is 300 years for the period 1976-2005, which causes a 30 percent reduction in end of 21st century record exceedance (see discussion with reviewer M.Sarofim for further details)."*

Other than that, most of my comments below are generally minor sentence edits.

**Line by line comments:**

Page 1, Line 17: I would also reference Fawcett et al. here as it has a good analysis of the probability of staying below various thresholds for the RCPs (based on simple models, not full GCMs, of course) (DOI: 10.1126/science.aad5761)

*Agreed.  Done.*

Page 2, Line 5: I am unclear how the authors determined that this commitment would lie "on the verge of economic and physical plausibility": as I read Smith et al., they evaluate the cost & capacity of negative emissions technologies, but do not define a plausible upper bound to how

fast society could implement these technologies if properly motivated. I would rephrase this sentence to be more judicious, e.g., "likely requiring substantial commitment to negative net carbon emission technologies in the 2nd half of the century".

*Rephrased as suggested*

Page 2, Line 19-20 and 23-24: it seems to me that "finding time periods from other scenarios" is equivalent to "another approach is to 'time-shift' by taking periods...": I would delete one of these descriptions.

*Agreed, removed the earlier sentence.*

Page 4, line 14: delete "in a decade" (it is redundant with "by 2027").

*Done.*

Page 5, line 2-3: I understand why the authors made the justifiable choice of using RCP8.5 non-GHG forcings for all the scenarios – however, it would be useful to have a brief aside that notes that a 1.5 degree scenario that is internally self-consistent might look slightly different than a 1.5 degree scenario that is a hybrid between RCP8.5 nonGHG forcings and low-GHG-concentrations. In particular, sulfur emissions might be C2 even lower in a 1.5 degree scenario (like RCP2.6 has lower sulfur than RCP8.5, though comparing RCPs should also be done with caution because they come from different IAMs so not all differences are necessarily due to policy effects), which would require even tighter GHG emissions reductions, but more relevantly for the paper, might also impact temperature patterns because aerosols have different land/ocean and hemispheric forcings than well-mixed GHGs. Similarly, a 1.5 degree future that relies heavily on bio-fuels would have very different land-albedo forcings. I also note that the choice of setting RCP2.6 as a limit to non-CO2 GHG reductions has an influence as well, possibly contributing to the necessity for the long-term CO2 emission floor to be negative (Table C1, column 7) because the CFCs and maybe N2O have lifetimes long enough that their concentrations would not have stabilized by 2200, requiring ongoing offsetting negative CO2 emissions (I think this study does not include SF6 or PFCs).

*Added the following paragraph to address these points:*

*It should be noted that the assumptions of RCP8.5 trajectories for non greenhouse gas forcers is implemented for practical reasons to make the present study tractable. However, a fully self consistent 1.5 degree scenario from an Integrated Assessment Model (IAM) would likely have slight differences. Sulfur emissions are lower in RCP2.6 than in RCP8.5, and maybe lower still in a 1.5 degree scenario (although there large differences in sulfur emissions between individual IAMs with the same policy constraints).*

*However, any given global temperature target could be achieved with different combinations of aerosol forcing and greenhouse gas forcing, but with regional differences in temperature and precipitation \citep{xu2015importance,pendergrass2015does}, and changes in land use necessary for large scale biofuel production would change surface albedo \citep{caiazzo2014quantifying}. An IAM could also have additional degrees of freedom, with the capacity to reduce N2O and CFC emissions below the RCP2.6 minimum levels.*

i would also potentially be curious regarding where CESM falls in terms of climate sensitivity in the larger CMIP universe, which would also determine how aggressive the GHG mitigation would need to be to stay below given targets.

*Added the following:*

*CESM also has a higher climate sensitivity (4.0K, \cite{gettelman2012evolution}) than the CMIP5 mean (3.6K, \cite{webb2013origins}), and so emissions need to be reduced faster than average for this model in order to meet any given temperature target.*

Page 5, line 3: Fix parentheses – in Kay et al. (2015).

*Done*

Page 5, line 18-20: I would suggest replacing the Rahmstorf (2007) semi-empirical approach with that from Kopp et al. (2016) (doi: 10.1073/pnas.1517056113): Rahmstorf is a co-author of the latter paper, which claims to reconcile the semi-empirical approach with process-based models, and therefore I would consider this to be an improved update to the Rahmstorf equations. (a correction would still be required for ice sheet melt, so Horton et al. 2014 might still be appropriate there).

*Done as suggested. Figure 1 has been recomputed with the Kopp semi-empirical model.*

Page 8, Line 15-16 (and Figure 3 generally): I note that during the period 1976-2005, models have already exceeded the maximum historical observed temperatures from 1976-2005. This makes me wonder whether the appropriate comparison should be between future model and historical model, rather than future model compared to historical observations. Either that, or there should be a discussion of this potential discrepancy. Reading Lehner et al., I think that paper did do compare to historical models – e.g., Figure 3 has both the observed 1920-2014 and the model 1920-2014, so that one can compare model-future to model-past, e.g., like-to-like. I recognize that there is still the potential for model bias to creep in here (as discussed in Lehner): if the model has more variability than the observed, then it is harder to exceed records, and vice versa, but I think a like-to-like is a cleaner comparison.

*As in our response to your major point, our issue here is which historical simulation to use - using all ensemble members to define records would create a meaningless result (because the records would be over a much longer time period). Using just one model would introduce an arbitrary bias. We have added a caveat paragraph to explain the potential for model bias - but we feel that the plot has most value in contrast to actual real world records, with a marked increase between the model's exceedance of real-world records in the late 20th century and the mid 21st century suggesting that the model biases do not dominate this metric.*

In addition, for section 3.3, I don't see a cited source for the observed temperatures – is it BEST as in Lehner? Or ERAi? I would suggest an additional paragraph here which could do several things: 1) note the source of observed temperature data,

*We already do - we are using the BEST data as in Lehner (2016):*

*"Observations are from \citealt{rohde2013berkeley}, as in \cite{lehner2016future}. "*

2) discuss how well the model reproduces the observations over the 1980-2005 period (and/or if any bias-correction is being used here),

*We have added the following caveat:*

*"It is notable that some records in Figure \ref{record} are exceeded before 2005 because the historical evolution of the CESM ensemble differs from the real-world historical evolution and there could potentially be some model regional model biases. "*

*But we also note that the IPCC found CESM to be one of the better models in representing historical temperature extremes, as follows:*

*"\cite{flatocoauthors}, Figure 9.37 shows that the model has one of the better simulations of extreme temperature and precipitation metrics in the CMIP5 archive. "*

3) discuss the model-observation comparison over the 2006-2016 period.

*Thanks for the suggestion - this period is, we think, an assurance that the biases in CESM are not dominating global or regional aggregate numbers of records exceeded:*

*"However, the behaviour of CESM in the period 2006-2016 is in within the range of model record exceedance (both globally and in each of the regions considered), giving confidence that regional biases are not strongly influencing this metric."*

Page 8, Line 26-27: Related, is this "noteworthy" statement regarding the 2006-2016 period? Please clarify, and see above suggestion.

*We have deleted this sentence.*

pg 8, line 33-34: 1) some regions experience a greater increase in extreme than in mean: is the opposite true as well? 2) following up on that: it would be very interesting to have a quantitative estimate of this effect: e.g., x% of the land area experiences a warming of extremes more than 50% faster than the mean, while y% experiences a warming 50% slower than the mean. Or, averaged across land areas, extremes warm x% faster than the mean. Or something like that.

*Done as suggested:*

*"(in 2.0degNE, 9\% of the land area experiences a warming of extremes more than 50\% faster than the mean, while less than 1\% experiences a warming 50\% slower than the mean)."*

pg 9, line 9: I might note that the greater signal to noise is seen at lower latitudes even though absolute warming at those latitudes is generally smaller (which has the opposite effect of there being less variability at those latitudes).

*Agreed, done:*

*"... the greatest signal to noise is seen at lower latitudes (although the absolute magnitude of warming is smaller)."*

pg. 21, line 5: stray period should be deleted.

*Fixed*

Figure 2: "subplot" should be singular.

*Fixed*

Figure 4: Is the historical period 1976-2005? Please specify. Also, it is based on observed (like Figure 3) or on modeled historical (like I think that most other figures do)

*Fixed:*

*"Maps showing the expected number of times that the modeled historical 1 in 20 year 3-day warm event in the period 1976-2005 would be exceeded during the period 2071-2100"*

Figure 6: legend needs more detail: I assume that black is modeled historical, but it could potentially be observed. Also, what's the time period of smoothing – annual?

*Expanded as follows:*

*"Changes in annual mean precipitation at the (a) Global, (b) Land-only and (c) high Northern latitude land. Values are relative to the 1921-1960 average. Grey lines show members of the historical CESM ensemble, while black line shows the historical mean. Thin colored lines show individual ensemble members for future scenarios, thick bold lines show the ensemble mean."*

Figure C2: Please include a legend for the colors as in Figure 1.

*Done*

Also, I don't think it that this figure extends far enough to demonstrate this, but I'd be curious about whether the 1.5NE and the 1.5OS could be used to investigate path dependence/memory/inertia. For example, one might expect some additional warming of the Arctic Ocean during the years in which the temperature is above 1.5 degrees which might take a number of years to dissipate even after the global surface air temperatures have returned to 1.5 degrees, which might lead to slightly lower sea ice extent in the 1.5OS case than then 1.5NE case for some years after stabilizing back at 1.5 degrees. (One could go further, and imagine hypothetical tipping points that could be exceeded in the 1.5 OS case which would not be resolved by cooling back down to 1.5 degrees, but I would imagine that this would be somewhat unlikely, and even if such a tipping point existed, this modeling system might not be able to catch it). (sea level could be a place where there might be some long-term memory of a brief excursion to 1.7 degrees, as in Zickfeld et al., doi: 10.1073/pnas.1612066114: Figure 1(c) seems to show this – it might be interesting to note the divergence between 1.5OS and 1.5NE at 2100 and what date the two scenarios become equal, if they ever do)

*Thanks for the suggestion.  Our co-author Alex Jahn is writing a dedicated paper on this topic, so we defer these questions of path dependency to that study.*

Table B1: Would it be possible to include an additional column with the values of each parameter that resulted from the calibration process?

*Done.*

**Response to reviewer 3 (KT Tanaka)**

1) I start with a broad comment related to the interpretation of the results. The paper ends with the statement stressing the differences in impacts between 1.5◦C to the 2◦C levels: "Irrespective of feasibility, these simulations indicate that a relaxation of ambition from the 1.5◦C to the 2◦C level would result in significantly greater impacts at the global scale, in the tropics and at high latitudes." The abstract also highlights the differences, rather than the similarities: "Exceedance of historical record temperature occurs with 60 percent greater frequency in the 2◦C climate than in a 1.5◦C climate aggregated globally, and with twice the frequency in equatorial and arid regions. Extreme precipitation intensity is statistically significantly higher in a 2.0◦C climate than a 1.5◦C climate in several regions. The model exhibits large differences in the Arctic which is ice-free with a frequency of 1 in 3 years in the 2.0◦C scenario, and only 1 in 40 years in the 1.5◦C scenario."

I take issue with the direction of argument, which is somewhat implicit in this paper. The paper makes me wonder what are the motivations. It is perhaps too broad to raise this here, but given the upcoming IPCC Special Report on Global Warming of 1.5◦C, are we as a community in charge of concluding urgently that there are discernable differences in impacts between 1.5 and 2◦C warming levels? The reason why I am raising this is that my overall impression of the results is drawn more toward the similarities. Visual inspection of the series of results certainly shows that there are significant (but not drastic, except for the sea ice (Fig. 1)) differences for various metrics (e.g. extreme precipitation (Fig. 10)) at the global mean level. But when it comes to regional and grid levels, differences are generally obscured by spatial and temporal variability as indicated by overlapping uncertainty ranges (just like any other global climate projections). In other words, similarities are more dominant than differences in my eyes. If there were multiple models performing the simulations, regional differences could be even less tantalizing. As a suggestion, I would think it is worth pointing out the similarities, not just the differences, at the abstract level. If the authors wish to bring forward only the differences, I would suggest that the basis of judgement be clarified to substantiate the claim.

*Thanks for this point, which is well taken. We have weakened the language of the abstract as follows:*

*"precipitation intensity is statistically significantly higher ... in some specific regions (but not all). " and*

*"Significance of impact differences with respect to multi-model variability is not assessed."*

*We have also added caveats to our concluding comments as follows:*

*"Irrespective of feasibility, these simulations indicate that a relaxation of ambition from the 1.5C to the 2C level would result in significantly greater impacts in some regions, at least compared with internal variability in CESM. Further study should consider these results in a multi-model context, using HAPPI and pattern scaling work together with these coupled single model experiments to produce a comprehensive assessment of avoided impacts in high mitigation scenarios."*

2) In my view, comparisons between 1.5degNE and 1.5degOS results are worthy of more discussion especially in the final section of the paper because it informs what the overshoot means in the context of 1.5◦C stabilization. It is unclear how the Paris Agreement would deal with an overshoot from the Agreement text. But, given the closing door for the 1.5◦C target as pointed out in this paper (page 15, line 5), possibilities of overshooting the target before achieving it are ever more relevant. As far as I am aware, implications of overshoot in the context of 1.5◦C target are not specifically analyzed in previous studies (e.g. (Rogelj et al. 2015)). I think a more dedicated discussion on the comparison between 1.5degNE and 1.5degOS results would thus be useful.

*Good point, thanks.*

*We've added the following to the Conclusions:*

*"Our study considered two mechanisms to achieve 1.5C, one which stabilizes by the mid 21st century, while the other overshoots reaching 1.7C in 2050 and stabilizes at 1.5C by 2100. Although the focus of the impact studies considered here have compared the equilibrium states at 1.5 and 2C, our scenarios allow a consideration of additional impacts which the overshoot would imply. In 2050, an additional 10 percent of global land area would be expected to exceed historical summer temperature records in the 1.5C overshoot, compared with the 1.5C stabilization case - although differences are not significant at the gridcell level. Our results do not suggest significant differences in sea level rise between the 1.5C overshoot case and the stabilization case and ice-free Arctic summers are simulated to be rare in both scenarios. Our analysis not suggest any evidence of long term climatic difference post-2100 of the overshoot relative to the stabilization case."*

3) Fig 1 shows that significantly negative CO2 emissions (about -2 GtC/yr in average) for more than 50 years (1.5degNE case) do not lead to a decline in the global-mean temperature. It is a removal of roughly 100 GtC from the atmosphere. I think this appears at odd with the rule of thumb that the stabilization level is determined by the cumulative CO2 emissions (Allen et al. 2009). Is there any explanation or perhaps some references that help clarify this temperature response?

*We're working on a paper to talk about this exact issue. But in short, we don't think that the Cumulative emissions rule of thumb is useful in a highly negative scenario. To show this convincingly requires a large ensemble of models with a range of carbon cycle responses - which we're in the process of performing.*

4) While the carbon in the land surface (as C sub l) is shown in Fig A1, it does not seem to be the case from the text that the land carbon cycle itself is explicitly modeled. Only the climate-land carbon cycle feedback is provided without being linked to the land carbon mass (Equation (A2)).

Furthermore, in many simple climate models, CO2 fertilization effect is modeled as a logarithmic function of the fractional increase of atmospheric CO2 concentration from preindustrial level (e.g. see equation (2.1.50) in page 28 of (Tanaka et al. 2007)). On the other hand, Equation

(A2) indicates that CO2 fertilization effect is not a function of atmospheric CO2 concentration. These points need to be clarified because applicable ranges of this model may be limited to low scenarios because of the treatment of carbon cycle-related feedbacks.

*It is correct that the land carbon cycle pool does not scale the carbon cycle feedback in this version of the model, but it does include a simple linear representation of CO2 fertilization, following Friedingstein 2003 (but as a prognostic model, rather than diagnostic). The ocean carbon model is simple, temperature dependent diffusive (i.e. not using the Friedingstein formulation. We add this caveat.*

*Note that the land carbon cycle feedback is not a function of the land carbon pool in this version, but is modeled using the parameters of \cite{friedlingstein2003positive}, where land carbon uptake is governed linearly by atmospheric CO2 content and temperature.*

5) The paper says in page 3 "Our main design choice was to minimize the number of its degrees of freedom to allow for fast calibration to reproduce the global mean trajectory of any given GCM." But when I look at the number of parameters, especially those for CH4 and N2O, I must say it is not really a model of minimal complexity. As some of the co-authors are aware, I developed a simple climate model (Tanaka et al. 2007; Tanaka et al. 2009), which I consider simple but not minimal at all. Even my model has less tunable parameters for CH4 and N2O (Table 3.2 of (Tanaka et al. 2007)). But this is just a naming issue, not a scientific one.

*The point is well taken. The non-CO2 portion of the model is based on a prior study, and we agree is not maximally simple. We have changed the sentence to simply read:*

*"We have provided a simple multi-gas climate model to perform this emulation,"*

Nevertheless, I do not understand some of the parameters in Table B1. For instance, the present-day growth rates for CH4 and N2O (ppb/a) and the present-day concentration of N2O should be model outputs, rather than model parameters because it is stated in page 17 lines 10-11: "The inputs to MiCES are global total emissions of greenhouse gas emissions (CO2, CH4, N2O, CFCs, HCFCs, CO)." This requires a clarification.

*The CH4 and N2O models are from Prather (2012), which allows for bias correction in present day growth rates. We've added the following to clarify this:*

*"The source code for MiCES is included in the supplementary material of this paper. Non-CO$_2$ forcings $F(t)$ are calculated using the atmospheric chemistry model defined and published in \cite{prather2012reactive}, which calculates the lifetimes and radiative forcings of non-CO$_2$ atmospheric components (CH$_4$, N$_2$O, HCFCs, CFCs), the model includes some bias correction for present day concentrations and growth rates. "*

*Prather, M. J., Holmes, C. D., and Hsu, J.: Reactive greenhouse gas scenarios: Systematic exploration of uncertainties and the role of atmospheric chemistry, Geophysical Research Letters, 39, 2012.*

Technical comments:

Appendix A The notation for the conversion factor between ocean carbon content in Pg and ocean carbon concentration is not consistent. It is rho in some places but rho sub o in other places.

*Fixed*

Page 15: Line 25 The sentence is unfinished.

*Fixed*

Page 15: Equation (A1) One of the brackets is not closed.

*Fixed*

Page 16: Line 18 Perhaps "due to" instead of "due"?

References Allen MR, Frame DJ, Huntingford C, Jones CD, Lowe JA, Meinshausen M, Meinshausen N (2009) Warming caused by cumulative carbon emissions towards the trillionth tonne. Nature 458 (7242):1163-1166. doi:10.1038/nature08019 Rogelj J, Luderer G, Pietzcker RC, Kriegler E, Schaeffer M, Krey V, Riahi K (2015) Energy system transformations for limiting end-of-century warming to below 1.5 [deg]C. C4 Nature Clim Change 5 (6):519-527.

doi:10.1038/nclimate2572 Tanaka K, Kriegler E, Bruckner T, Hooss G, Knorr W, Raddatz T (2007) Aggregated Carbon Cycle, Atmospheric Chemistry, and Climate Model (ACC2) – description of the forward and inverse modes. Reports on Earth System Science, vol 40. Max Planck Institute for Meteorology, Hamburg Tanaka K, Raddatz T, O'Neill BC, Reick CH (2009) Insufficient forcing uncertainty underestimates the risk of high climate sensitivity. Geophys Res Lett 36 (16):L16709. doi:10.1029/2009gl039642 Interactive comment on Earth Syst. Dynam. Discuss., doi:10.5194/esd-2017-42, 2017. C5

**List of substantial changes**

- Abstract revised and further caveats added on scope of paper with respect to other literature on 2 degree and 1.5 degree targets.
- Further citations added to introduction
- Statements of socio-economic plausibility have been caveated.
- Expanded discussion of the benefits of the HAPPI protocol
- Added discussion of the suitability and performance of CESM1-CAM5 in terms of the variables considered in the paper.
- Added discussion on the choices of non-CO2 forcers in the emissions trajectories
- Added caveats on the (untested) long term evolution of the scenarios in the fully coupled model.
- Added discussion on the use of real-world temperature records in FIgure 3, rather than model historical generated records
- Removed 3 paragraphs from conclusions, so results are no longer repeated from the previous section.
- Reworded conclusions and added paragraph to note that for some regions, especially at the point level, there are no significant differences between 2 and 1.5 degree climates.
- Reworded conclusions to illustrate relative benefits of other approaches (especially the HAPPI protocol).
- Clarified land carbon feedback in appendix A
- Clarified use of bias correction in N20 and CH4 models
- Added calibrated model parameter column to table B1

[revised manuscript text omitted]

Fawcett, A. A., Iyer, G. C., Clarke, L. E., Edmonds, J. A., Hultman, N. E., McJeon, H. C., Rogelj, J., Schuler, R., Alsalam, J., Asrar, G. R., et al.: Can Paris pledges avert severe climate change?, Science, 350, 1168–1169, 2015.

Fischer, E. M. and Knutti, R.: Anthropogenic contribution to global occurrence of heavy-precipitation and high-temperature extremes, Nature Climate Change, 5, 560–564, 2015.

Flato, G.: Coauthors, 2013: Evaluation of climate models. Climate Change 2013: The Physical Science Basis, TF Stocker et al., Eds, 2013.

Friedlingstein, P., DUFRESNE, J.-L., Cox, P., and Rayner, P.: How positive is the feedback between climate change and the carbon cycle?, Tellus B, 55, 692–700, 2003.

Fu, Q. and Feng, S.: Responses of terrestrial aridity to global warming, Journal of Geophysical Research: Atmospheres, 119, 7863–7875, 2014.

Gasparrini, A., Guo, Y., Hashizume, M., Lavigne, E., Tobias, A., Zanobetti, A., Schwartz, J. D., Leone, M., Michelozzi, P., Kan, H., et al.: Changes in susceptibility to heat during the summer: a multicountry analysis, American journal of epidemiology, 183, 1027–1036, 2016.

Gettelman, A., Kay, J., and Shell, K.: The evolution of climate sensitivity and climate feedbacks in the Community Atmosphere Model, Journal of Climate, 25, 1453–1469, 2012.

Giorgi, F. and Mearns, L. O.: Calculation of average, uncertainty range, and reliability of regional climate changes from AOGCM simulations via the "reliability ensemble averaging"(REA) method, Journal of Climate, 15, 1141–1158, 2002.

Guo, Y., Gasparrini, A., Armstrong, B., Li, S., Tawatsupa, B., Tobias, A., Lavigne, E., Coelho, M. d. S. Z. S., Leone, M., Pan, X., et al.: Global variation in the effects of ambient temperature on mortality: a systematic evaluation, Epidemiology (Cambridge, Mass.), 25, 781, 2014.

Herger, N., Sanderson, B. M., and Knutti, R.: Improved pattern scaling approaches for the use in climate impact studies, Geophysical Research Letters, 42, 3486–3494, 2015.

Horton, B. P., Rahmstorf, S., Engelhart, S. E., and Kemp, A. C.: Expert assessment of sea-level rise by AD 2100 and AD 2300, Quaternary Science Reviews, 84, 1–6, 2014.

[revised manuscript text omitted]